# The impact of future changes in climate variables and groundwater abstraction on basin-scale groundwater availability

Steven Reinaldo Rusli[1,2], Victor F Bense[1], Syed M.T Mustafa[1], and Albrecht H Weerts[1,3]

[1]Hydrology and Environmental Hydraulics, Wageningen University, 6708 PB Wageningen, The Netherlands
[2]Civil Engineering Department, Faculty of Engineering, Parahyangan Catholic University, Bandung 40141, Indonesia
[3]Operational Water Management, Department of Inland Water Systems, Deltares, The Netherlands

**Correspondence:** Steven Reinaldo Rusli (steven.reinaldo.rusli@gmail.com)

**Abstract.** Groundwater is under the pressure of changing climate and increasing anthropogenic demand. In this study, we project the effect of these two processes on the projected future groundwater status. Climate projections of Representative Concentration Pathway (RCP) 4.5 and RCP8.5 from the Coupled Model Intercomparison Project Phase 6 (CMIP6) drive a one-way coupled fully distributed hydrological and groundwater model. In addition, three plausible groundwater abstraction scenarios with diverging predictions from increasing, constant, to decreasing volumes and spatial distribution are used. Groundwater status projections are assessed for the short-term (2030), mid-term (2050), and long-term (2100) periods. We use the Bandung groundwater basin as our study case, located 120 km from the current capital city of Indonesia, Jakarta, which is currently under a relocation plan. It is selected as the future anthropogenic uncertainties in the basin, related to the projected groundwater abstraction, is in agreement with our developed scenarios. Results show that changes in the projected climate input, including intensifying rainfall and rising temperature, do not propagate notable changes in groundwater recharge. Under the current unsustainable groundwater abstraction rate, the confined piezometric heads are projected to drop up to a maximum of 7.14 m, 15.25 m, and 29.51 m in 2030, 2050, and 2100, respectively. When groundwater abstraction expands in proportion to the present population growth, the impact is worsened almost two-fold. In contrast, if the groundwater abstraction decreases because of the relocated capital city, the groundwater storage starts to show replenishment potential. As a whole, projected groundwater status changes are dominated by anthropogenic activity, and less so by changes in climatic forcing. The results of this study are expected to demonstrate and inform responsible parties in operational water management on the issue of the impact of projected climate forcing and anthropogenic activity on future groundwater status.

## 1 Introduction

Groundwater, as one of the major sources of water on earth, has been known to be over-exploited in many basins worldwide (Bierkens and Wada, 2019; Gleeson et al., 2020), which has further caused global depletion of groundwater resources (Wada et al., 2010). In more than half of the sub-districts located in the northwestern part of Bangladesh, the estimated groundwater abstraction has a higher volume than the simulated groundwater recharge (Shahid et al., 2015), and overexploitation of groundwater for irrigation, identified as the primary factor, contributes to the decline in groundwater level in these areas (Mustafa et al., 2017). Even more drastically, twenty-one (out of twenty-three) provinces in China were diagnosed with groundwater

over-exploitation-related problems (Lili et al., 2020). In the northeastern part of Brazil, the intensification of groundwater exploitation has caused piezometric surface drawdowns of up to 100 m (de Luna et al., 2017). In the archipelago part of Spain, Gran Canaria, the currently accumulated groundwater depletion would require a few decades to recover (Custodio et al., 2016). From all such cases, we can see the severe impact driven by anthropogenic activities through groundwater abstraction on the groundwater regime. Dwindling groundwater tables, depleting groundwater storage, and degrading groundwater quality, have led to various consequences such as land subsidence (Chen et al., 2022), wetland deterioration (Mancuso et al., 2020), groundwater pollution (Meng et al., 2022), and seawater intrusion (Momejian et al., 2019).

Besides anthropogenic activities such as groundwater abstraction discussed above, climatic variability may also play an important role in a changing groundwater regime since groundwater recharge is the dominant driver of groundwater flow. Surface and soil properties aside, groundwater recharge is modulated by precipitation, temperature, radiation, and other climate variables. The recent changes in these variables' patterns, frequencies, and extremes, therefore, have led to the alteration of the groundwater table distribution. Several studies suggest that changes in climate can contribute positively to an increase in groundwater recharge (Tillman et al., 2016; Patle et al., 2018; Gaye and Tindimugaya, 2019). This occurs when rainfall intensification provides a higher volume of water, thus providing an opportunity for more abundant groundwater resources. Some others advocate contrarily and estimate a decline in groundwater recharge as a consequence of climate change (Pardo-Igúzquiza et al., 2019; Anurag and Ng, 2022; Trásy-Havril et al., 2022). Mostly, such a reduction is related to the higher potential evapotranspiration driven by the increasing temperature. Some others found the trend to be less definitive, varying per case depending on various factors, and involving large uncertainties in its quantification (Meixner et al., 2016; Smerdon, 2017; Yawson et al., 2019; Wu et al., 2020b; Wang et al., 2021).

As much as both anthropogenic and climatic factors have influenced the groundwater regime in the past centuries even at the global scale (Döll et al., 2012), they also, to various extents, control the current and future status of the subsurface resource (Stevenazzi et al., 2017; Liu et al., 2022). Therefore, future groundwater resource prediction relies greatly on climate projections and anthropogenic scenarios. However, the degree to which these two factors influence the groundwater resource varies in each case. Some studies suggest that changes in groundwater abstraction (anthropogenic factors) are more influential to the groundwater level compared to changes driven by climatic factors (Varouchakis et al., 2015; Brewington et al., 2019; Mustafa et al., 2019). On the other hand, Davamani et al. (2024) compiled a list of studies that propose otherwise. Within the list, it is shown that groundwater recharge will drop significantly under the impact of climate change (Olarinoye et al., 2020; Soundala and Saraphirom, 2022). This signals the uncertainties and the high spatial variability in terms of the impact of both climatic and anthropogenic factors on groundwater recharge, and further the groundwater level and availability. Therefore, studies encompassing various spatial scales - global, regional, and even local ones - on the influences of these factors are important to be addressed (Jyrkama and Sykes, 2007; Hughes et al., 2021).

Regarding climate projection studies, the Coupled Model Intercomparison Project (CMIP) takes an important position in coordinating the global climate models (GCMs) worldwide. In its current sixth development phase, CMIP6 (Eyring et al., 2016) distributes climate model outputs from numerous GCMs run by various model groups under different Representative Concentration Pathway (RCP) scenarios (IPCC, 2021), a set of pathways developed specifically on the span of projected

radiative forcing values by the year 2100 (van Vuuren et al., 2011). With numerous hydrological forcing projections available in multiple scenarios, it is possible to simulate future groundwater recharge using hydrological models, from catchment to global scales (Yuan et al., 2015; Zhao et al., 2021; Hua et al., 2022).

While climate variables have only partial repercussions on groundwater recharge as it is also controlled by other factors, in particular the basin surface and subsurface properties, groundwater abstraction directly removes the groundwater from subsurface storage. Regarding anthropogenic projection scenarios, many studies develop scenarios in which the groundwater abstraction rates increase (Chang et al., 2020; Ansari et al., 2021; Aslam et al., 2022), in line with rising populations. Only a few studies projected the abstraction to decrease in the future (Mustafa et al., 2019; Siarkos et al., 2021), and when they do, it is estimated not as the likely scenario but as a recommended policy to achieve sustainable abstraction rates. Nevertheless, in some specific basin-scale areas, decreasing future groundwater abstraction might be a real possibility, and hence is not less important to be studied in comparison to the increasing abstraction scenario. This is one of the novelties offered in this study, along with the coupled modelling approach and the assessment of shallow and deep groundwater availability.

In this study, we aim to envisage future groundwater availability under several climatic and anthropogenic scenarios, especially considering the spatially volatile variability of the influences of these factors on groundwater resources. We test the approach on the Bandung groundwater basin in Java, Indonesia. This would contribute novel findings on the degree of influence of both climatic and anthropogenic factors in such a highly groundwater-dependent and tropical area. While currently developed in a rising population trajectory, the Indonesia capital city relocation plan could, in reverse, steer the future groundwater abstraction down in the Bandung groundwater basin. This is a rather unique situation, as there are not many basins in the world that are seriously facing the possibility of future reduced groundwater abstraction. The Bandung groundwater basin and the city of Jakarta are closely connected. The urban and industrial sector development in the study area is highly influenced by the demographic and socio-economic activity within and around the capital city. With the plan to relocate the capital to Borneo Island, it is predicted that many aspects of the study area would be impacted, including the decrease in groundwater abstraction volume, rate, and spatial distribution. However, there have been several challenges in the early phase of the relocation that might lead to further repercussions. Should any consequences arise on the project such as delay, postponement, or even worse, revocation, the groundwater abstraction in the Bandung groundwater basin would still be projected to increase. Such uncertain circumstances of future groundwater abstraction are reflected by the developed anthropogenic scenarios in this study.

Under the future climatic forcing and groundwater abstraction uncertainties, in our analysis, we simulate the groundwater level and storage changes using a one-way coupled distributed hydrological model (Wflow_sbm) and groundwater flow model (MODFLOW) (Rusli et al., 2023a, b). By applying multiple climatic forcing and abstraction scenarios, we aim to specifically (1) quantify the impact of future climate projection on groundwater recharge, and (2) assess the impact of the changing groundwater abstraction on groundwater status in the study area. It is expected that the outcome of this study will be useful in understanding basin-scale future groundwater availability as well as the controlling factors and its processes. We also believe that this study would provide valuable input for the Bandung groundwater basin authorities to improve the current and future groundwater policy and management.

## 2 Materials and methods

### 2.1 Study area

#### 2.1.1 Hydrological situation, hydrogeological setting, and the current state of the art

The Bandung groundwater basin is located in the western part of Java Island, Indonesia, close to the current capital city, Jakarta (see Figure 1a). It covers a total area of over 1,699 km$^2$ and was populated by approximately 10 million people in 2020. Topographically, it is surrounded by steep mountains around its perimeter but is vastly plain in its middle part, where the urban areas are developing. Its elevation ranges between 640 m and 2,500 m above sea level. Its average annual rainfall between 2005 and 2018 is estimated between 1,970 and 2,850 mm per year (Rusli et al., 2021). The average discharge at the surface catchment outlet during the same period is observed at 73.86 m$^3$ s$^{-1}$, but with a large temporal variation, with the highest and lowest recorded discharge of 469.29 m$^3$ s$^{-1}$ and 4.4 m$^3$ s$^{-1}$, respectively. Our previous study also reported the spatial distribution and the pumping volume of the groundwater abstraction within the basin boundary (Rusli et al., 2023a). In summary, the water demand from the domestic and industrial sectors has been growing at a swift rate of up to 1.6 times increase over those 14 years. On average, the volume of groundwater abstraction from the upper and the lower aquifer is estimated at 122 million and 255 million m$^3$ per year, respectively. The 'upper' and 'lower' aquifers are discerned, as hydrogeologically the Bandung groundwater basin consists of multiple subsurface layers (see the conceptual groundwater flow model in Figure 1b, taken from (Rusli et al., 2023b)). Although the main aquifer is formed by a solitary geological formation named the Cibeureum Formation, it is interspersed in many locations by thin clay layers known as the Kosambi Formation. This aquitard zone conceptually divides the aquifer into two stratifications, hence the 'upper' and the 'lower' aquifer. Detailed data on the basin climate, hydrology, and hydrogeological features are presented in our previous studies (Rusli et al., 2021, 2023a, b).

According to previous studies, the groundwater level in the Bandung groundwater basin has been decreasing in the last three decades. The groundwater table dwindles, on average, from less than one meter per year around urban areas close to rivers and streams, to 2.45 m per year around the industrial area (Abidin et al., 2013; Gumilar et al., 2015). A correlation between the groundwater abstraction location and the dwindling groundwater table is visually apparent; the drop in groundwater level is distinctly higher in areas where the groundwater is abstracted from both the upper and the lower aquifer (Rusli et al., 2023b). In recent years, the rising pressure of water demand still increases groundwater abstraction, while the intensified rainfall pattern fluctuates the groundwater recharge. Furthermore, this is worsened by the fact that groundwater abstraction from the bottom aquifer is partially compensated by the loss of groundwater in the upper aquifer through vertical flow, which negatively impacts the unconfined groundwater table's disproportionate drawdown (Rusli et al., 2023b). All of the mentioned features of groundwater flow were simulated in our previous studies (Rusli et al., 2023a, b). Not only the characteristics of the subsurface flow are numerically reconstructed, but the one-way coupled hydrological-groundwater flow model is also supplemented, considering the limited number of calibration data, by additional validation of groundwater storage change using satellite gravimetry-based estimates (GRACE) (Rusli et al., 2023a) and semi-quantitative model evaluation using environmental water tracer data (Rusli et al., 2023b).

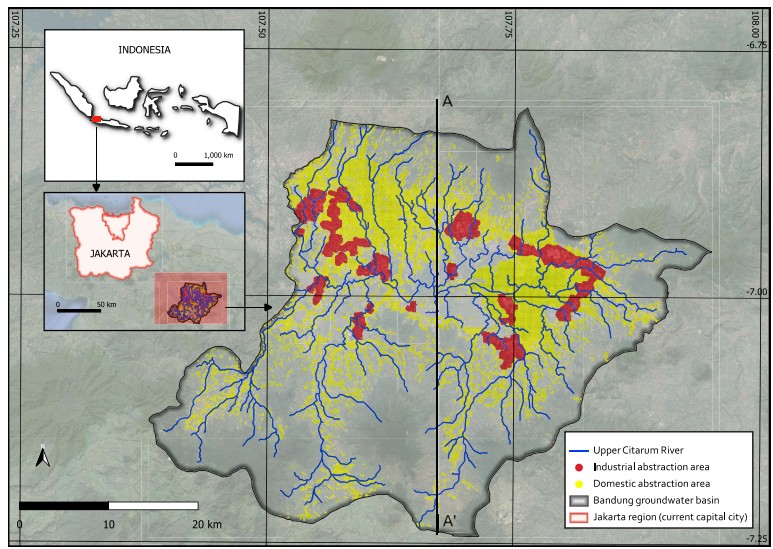

(a) The geographical location of the study area. The yellow and red area denotes the domestic and industry groundwater abstraction areas respectively. The study area's relative position is highlighted in the locator map in the top left corner, indicated by the red square in Java Island. The second locator map just below the main locator map highlights the close distance between our study area of the Bandung groundwater basin and Jakarta, the current capital city of Indonesia. This image is adapted from Rusli et al. (2023b).

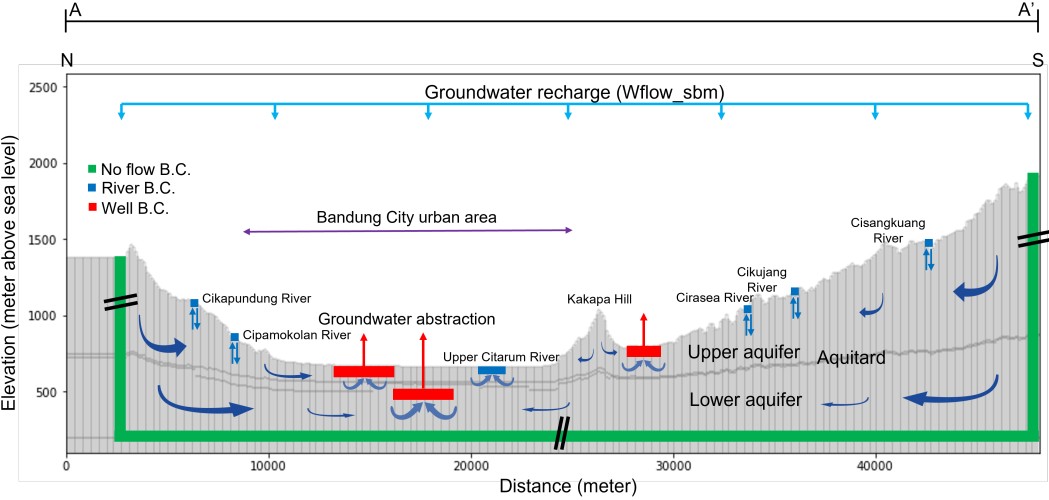

(b) The conceptual model of the Bandung groundwater basin from the north-south cross-section. Interspersing aquitard layers are found in between the upper and lower aquifers, interpolated based on the available borehole data. The figure also includes groundwater fluxes entering, leaving, and flowing within the domain, as well as the boundary conditions. This image is taken from Rusli et al. (2023b).

**Figure 1.** The overview of the Bandung groundwater basin as the study area.

### 2.1.2 The influence of Jakarta and the new capital city plan on the Bandung groundwater basin

Jakarta is a metropolitan city, with an area of 661.5 km$^2$, and a population density of 13,000 people per km$^2$. Its area extends to a larger Jakarta Metropolitan Area (JMA), with a total area of 4,384 km$^2$. In 2019, the number of daily commuters entering JMA was reported at 3.2 million people (Martinez and Masron, 2020). Undoubtedly, Bandung City, the largest city within the Bandung groundwater basin, is one of the cities with tangible mutual dependencies to Jakarta due to its neighboring distance (see Figure 1a). Bandung City was populated by 2.7 million people, while the greater area of the Bandung groundwater basin had a total population of 10.5 million people, both in 2023. The flows of demographic and socio-economic activities between these two cities make them often referred to as the Jakarta-Bandung mega-urban region (Pravitasari et al., 2018).

The future of the groundwater regime in the Bandung groundwater basin becomes uncertain with the latest geospatial planning of Indonesia. It is to move its capital city from Jakarta to Nusantara, located on even a separate island in Borneo (Nugroho, 2020; Mutaqin et al., 2021; Hackbarth and de Vries, 2021). Under the current schedule where the capital city will begin to be relocated in 2024 (de Vries and Schrey, 2022), the urban and industrial development in cities surrounding Jakarta, including those in the Bandung groundwater basin, is predicted to be impacted, as the relocation would not only move the center of government but also mobilize part of the residents (Kodir et al., 2021). Regulated in the Presidential Regulation of Indonesia number 63, 2022 (Indonesia, 2022), the relocation is scheduled to be finalized in 2045, with the first two phases' end date in 2030. By 2030, up to 750,000 people are predicted to be fully relocated from JMA and its surroundings, including the Bandung groundwater basin, to Nusantara City.

Considering the capital city relocation plan and the close affinity between our study area and Jakarta, it is reasonable to imagine that the former issue would dampen the growth of the urban and industrial area in the Bandung groundwater basin. Indirectly, it would also be possible to forecast that the pressure of pumping groundwater to fulfill the water demand in the future is possibly reduced. Having said that, such a trend did not always happen in other former capital cities, such as Rio de Janeiro (Silva Jr and Pizani, 2003), Lagos (Healy et al., 2020), and Yangon (Hashimoto et al., 2022). In these three ex-capital cities, the economic growth carries on, despite being at different rates, and is translated to increasing groundwater abstraction. The uncertainties involving the future water resources, specifically groundwater abstraction, in the Bandung groundwater basin are highly unsettling, thus multiple and diverging scenarios are necessary to be explored. Therefore, while the common conceptual understanding of groundwater abstraction projection is to increase in the future, in this study we define three scenarios with wide ranges, stretching from an increasing to decreasing future groundwater abstraction, described in Section 2.3.2.

### 2.2 Simulation workflow and temporal framework

There are two numerical simulations involved in the modeling framework within this study: (1) surface hydrological and (2) groundwater flow. The hydrological simulation is performed using the Wflow_sbm model (Section 2.4.2), with two climate variables as its main forcing: the rainfall and the potential evapotranspiration. This produces two outputs: the simulated river discharge and the groundwater recharge. The former is compared with observation data to evaluate the performance of the hydrological Wflow_sbm model simulation, while the latter is used to force the one-way coupled MODFLOW groundwater

flow model (Section 2.4.3). Aside from having the simulated groundwater recharge as its input, the groundwater flow model is also regulated by boundary conditions of groundwater abstraction. The outputs of the groundwater flow model include transient groundwater tables for the unconfined layer and piezometric heads for the confined layer, which allow to calculate the associated groundwater storage changes.

We consider the described simulation setup in this study in two phases based on the temporal categorization: the baseline and the future period. The baseline period is used as the 'control' of changes in the latter period; this is very important to note especially for variables whose projected changes are expressed in percentage relative to the benchmark. For the hydrological simulation using Wflow_sbm and the groundwater flow model using MODFLOW, the benchmark period is set between 2005 and 2015. The baseline hydrological simulation was forced by the CHIRPS rainfall estimates (Funk et al., 2015), and the groundwater abstraction defined in the groundwater flow model was estimated upon population number (Rusli et al., 2021). For the climate data temporal classification, the baseline period's temporal coverage starts from 1981 to 2015 following the categorization on the Copernicus Climate Data Store of MRI-ESM2-0 model group (Copernicus Climate Change Service, 2021). For the future period, we classify the temporal setting into three categories: the short-term future (up to 2030), the mid-term future (up to 2050), and the long-term future (up to 2100). These temporal classifications serve not only as the analysis checkpoints but also as the milestone for the projected groundwater abstraction spatial distribution, described in Section 2.3.2. On the whole, Figure 2 outlines the role of the temporal setup in this study (in the horizontal direction), from the baseline (left box) to the future (right box) period, as well as the setup for subsequent hydrological and groundwater flow simulations (in the vertical direction) in each period.

## 2.3   Future scenario development

In this study, we are going to independently develop two climatic scenarios (Section 2.3.1) and three anthropogenic scenarios (Section 2.3.2). The climatic scenarios involve changes in projected rainfall and potential evapotranspiration, influenced by temperature and radiation. The anthropogenic scenarios involve changes in not only the groundwater abstraction rate but also the spatial distribution of the pumping activities. These scenarios result in six combinations of outcomes that are analyzed, compared, and discussed in-depth in further sections.

### 2.3.1   Climate projection data and scenario

For the future period hydrological simulation, we use the CMIP6 (Coupled Model Intercomparison Project Phase 6) climate model runs (Eyring et al., 2016) under two greenhouse gas concentration trajectory scenarios: RCP (Representative Concentration Pathway) 4.5 and 8.5 (IPCC, 2021). The RCP4.5 is selected as the intermediate scenario, while the RCP8.5 is the extreme one. The considered variables are those required as the input to the Wflow_sbm model, specifically precipitation and near-surface air temperature as well as surface downwelling shortwave radiation and top-of-atmosphere (TOA) incident shortwave radiation. The latter three variables are used to estimate potential evapotranspiration using the method proposed by de Bruin et al. (2016). All the mentioned climate model products are publicly available on the Copernicus Climate Data Store (Copernicus Climate Change Service, 2021). Consistent with Section 2.2 and Figure 2, we use the data on both the baseline and future

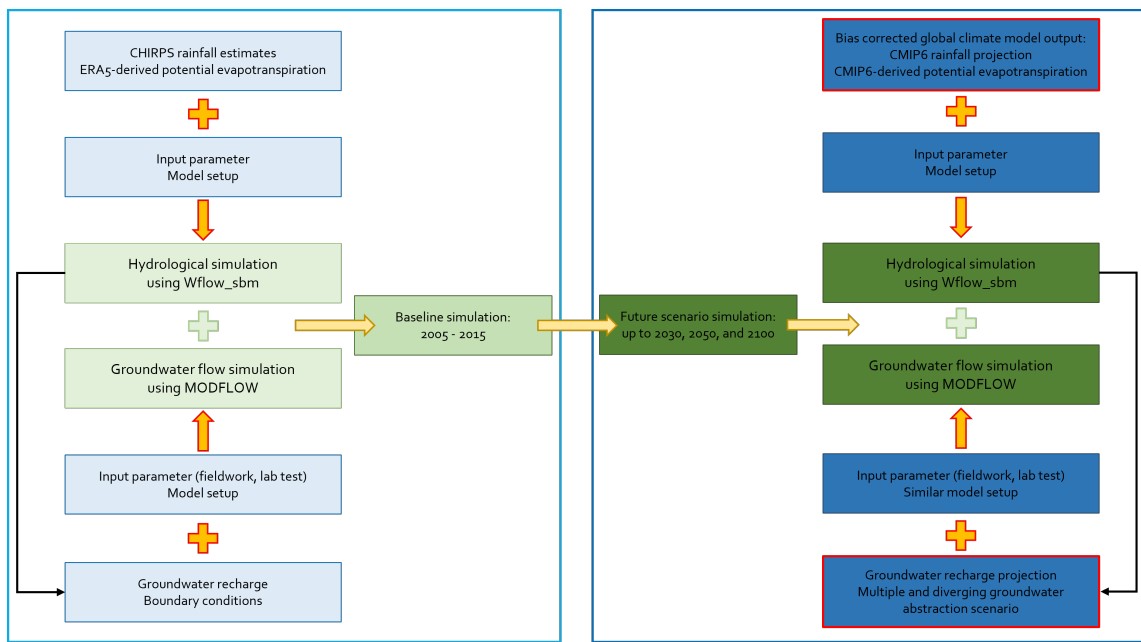

**Figure 2.** The overview of the modeling workflow. The left and right boxes show, respectively, the model setup during the baseline period and the future period. While both the hydrological and groundwater flow model's setup is similar during the two periods, the forcing data and boundary conditions are different and determined according to their respective configuration.

scenarios temporal framework; up to 2015 for the baseline period and up to 2100 for the future period. The used data have daily temporal resolution, following the daily resolution for the hydrological Wflow_sbm model simulation, and aggregated to
monthly resolution for the groundwater MODFLOW model simulation.

To select the model group, we apply criteria that the product should have (a) a spatial resolution of approximately $1°×1°$ considering the size of the study area, (b) a daily temporal resolution for rainfall and temperature following the hydrological model setup, and (c) a projection scenario of historical, RCP4.5, and RCP8.5; applicable to the four mentioned variables. Considering the criteria and other climate projection studies, we use the climate model outputs from the MRI-ESM2-0 model
group (Yukimoto et al., 2019) for the precipitation and near-surface air temperature. It has been suggested to perform well in CMIP6 outputs (Oruc, 2022), especially those related to cloud-related processes (Kawai et al., 2019). It has also been indicated to be one of the best-performing models in other studies, proven through multi-metric assessments, including in Southeast Asia region (Iqbal et al., 2021; Baghel et al., 2022), which is very close to our study area. It is not used for the global radiation data, however, as it does not cover the TOA incident shortwave radiation projection for the RCP 8.5. Therefore, for the global
radiation data, we use the climate model output from the GFDL-ESM4 model group (Krasting et al., 2018).

### 2.3.2 Groundwater abstraction scenarios

For the groundwater abstraction projection, we consider three diverging scenarios where the groundwater abstraction (a) increases, (b) stays constant, and (c) decreases in the future. Meanwhile, the groundwater abstraction during the baseline period is set according to estimates from our previous studies (Rusli et al., 2023a), increasing annually from 300 $Mm^3$ per year in 2005 to 495 $Mm^3$ per year in 2020. For all boundary conditions, the groundwater abstraction is distributed horizontally based on land use and vertically based on domestic/industrial water demand classification.

In this study, we propose a new approach to establishing the scenario where the future groundwater abstraction increases (scenario one). Commonly, the projected groundwater abstraction rate increases in proportion to the projected population growth. We indeed apply such a method to estimate the annual volume of the future groundwater abstraction in the study area, using an annual population growth rate of 1.36%, shown in Figure 3a. Under this scheme, the annual groundwater abstraction volume in 2030, 2050, and 2100 is projected to be 529, 839, and 1,346 $Mm^3$ per year, respectively. Considering the currently high population density in the Bandung groundwater basin, it is only logical that a surge in groundwater abstraction volume is accompanied by an enlargement of the abstraction area. In this paper, we expand the groundwater abstraction location by increasing the area of the initial abstraction location proportionally to the volume of abstraction.

The other two groundwater abstraction scenarios are based on Indonesia's capital city relocation plan. In the 'stay constant' scenario (scenario two), we assume that the urban and industrial area that are currently settling will continue to remain where they are now, with linear development (Figure 3b). It is also possible that there would be tangible development, but its impact will be compensated with technological advancement, for example, increasing efficiency or reducing losses of (groundwater) use and distribution. Therefore, the groundwater abstraction rate is set constant from 2020 to 2100. In the decreasing groundwater abstraction scenario (scenario three), we assume that the capital city relocation would decrease the population of the Bandung groundwater basin and water demand gradually in the future (Figure 3c). By 2100, it is assumed that the population would have halved from 2020, therefore also decreasing the groundwater abstraction rate by 50% to 248 $Mm^3$ per year, linearly interpolated. In scenarios two and three, the spatial distribution of the groundwater abstraction area remains the same as applied in the baseline period.

## 2.4 One-way coupled model simulation setup

### 2.4.1 Bias adjustment and statistical downscaling method of ISIMIP3b

In climate-related research, it is common to apply bias correction to climate simulation data as they generally have different statistical attributes to climate observation data (Lange, 2019). The discrepancies occur due to various factors, such as differences in spatial resolution and systematic biases. Therefore, bias correction, involving two steps method of bias adjustment and statistical downscaling, is necessary to bridge and minimize this gap. In this study, we apply the method tailored to the Inter-Sectoral Impact Model Intercomparison Project phase 3b (ISIMIP3b) for our bias correction (Lange, 2019, 2021).

In ISIMIP3b, the required data to be specified as the benchmark are the (a) historical 'ground truth' and (b) high-resolution data for the bias adjustment and statistical downscaling, respectively. In this study, we use CHIRPS estimates (Funk et al.,

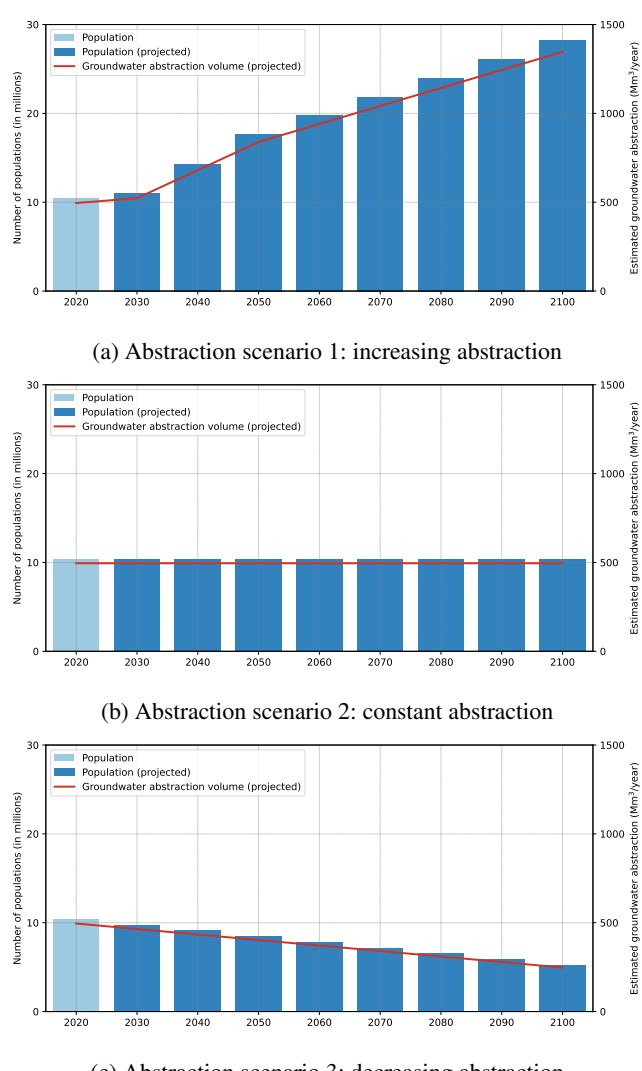

(a) Abstraction scenario 1: increasing abstraction

(b) Abstraction scenario 2: constant abstraction

(c) Abstraction scenario 3: decreasing abstraction

**Figure 3.** The three diverging scenarios of estimated groundwater abstraction volumes (right axis, Mm$^3$/year) based on the projected number of population (left axis, millions).

2015), with a spatial resolution of $0.25° \times 0.25°$, as the historical 'ground truth' rainfall estimates. The MRI-ESM2-0 model
output is available in lower spatial resolution than the CHIRPS estimates, therefore they first are re-gridded and resampled
to match the CHIRPS' spatial resolution, before the bias adjustment was applied. As CHIRPS is available at an even higher
resolution of $0.05° \times 0.05°$, we use it further as the benchmark for the statistical downscaling. The historical 'ground truth'
estimates for the other variables (near-surface air temperature, surface downwelling shortwave radiation, and TOA incident
shortwave radiation) are based on ERA5-Land hourly data (Copernicus Climate Change Service, 2019). All the datasets used
for the application of ISIMIP3b in this paper are listed in Table 1.

**Table 1.** Datasets used for bias adjustment and statistical downscaling of the climate model output

| Variables | Climate model output | 'Ground truth' data |
| --- | --- | --- |
| Precipitation | MRI-ESM2-0 | CHIRPS[1] |
| Near-surface air temperature | MRI-ESM2-0 | ERA5-Land |
| Surface downwelling shortwave radiation | GFDL-ESM4 | ERA5-Land |
| TOA incident shortwave radiation | GFDL-ESM4 | ERA5-Land |

[1] two CHIRPS products of different resolutions of $0.25° \times 0.25°$ and $0.05° \times 0.05°$ are used

### 2.4.2 Wflow_sbm model setup

The Wflow_sbm model (van Verseveld et al., 2022) is used to perform the hydrological simulation. With model parameters that mostly represent physical characteristics, using Wflow_sbm makes it easier to intuitively interpret and correlate the parameter values with physical catchment properties. In the last decade, Wflow_sbm has been widely used in hydrological modeling studies (López et al., 2016; Hassaballah et al., 2017; Gebremicael et al., 2019), including those in tropical regions in South East Asia (Wannasin et al., 2021; Rusli et al., 2021), delivering good performance, shown by KGE, NSE, and RMSE metric.

We use the same Wflow_sbm model as in our previous studies (Rusli et al., 2023a, b). Starting with high-resolution model parameterization based upon point-scale (pedo)transfer functions (PTFs) (Imhoff et al., 2020), it is followed by downscaling designated to the model resolution of $0.008° \times 0.008°$. We use (a) SoilGrids database (Hengl et al., 2017) for soil-related parameters estimation, (b) Monthly Leaf Area Index climatology for daily interception calculation (Gash, 1979), (c) MERIT-DEM dataset (Yamazaki et al., 2017) for river network delineation (Eilander et al., 2020), and (d) vito land use map (Buchhorn et al., 2020) for deriving land-use related parameters. After the simulation of the surface processes, the infiltrated water flow is controlled by the $MaxLeakage$ parameter. Such a parameter influences the amount of water flowing from the pseudo water table to the deep groundwater (van Verseveld et al., 2024), hence groundwater recharge. It is usually only used for linking to a dedicated groundwater model, representing the water that is 'lost' to the model. Normally set to zero in all other cases, when the MaxLeakage is defined as higher than zero, the simulated water is treated to be lost from the saturated zone and runs out of the model. As the Wflow_sbm model only considers the first couple of meters of soil below the surface level, the water that leaves the saturated zone is then treated as the groundwater recharge. To calibrate the $MaxLeakage$ parameter, considering its importance, we optimized the KGE value between the observed and the simulated river discharge (Rusli et al., 2023a).

Precipitation and potential evapotranspiration are the primary forcing data to run the Wflow_sbm. Following the temporal setting, we prepare and split the forcing data into the baseline and future projection periods. For the baseline period, CHIRPS rainfall estimates (Funk et al., 2015) and potential evapotranspiration, derived from ERA5 temperature and global radiation data using the method from de Bruin et al. (2016), are used. Using the Extended Triple Collocation method (McColl et al., 2014), CHIRPS products were found to perform well in estimating rainfall in the study area in our previous study (Rusli et al., 2021). For the future period, the forcing data are obtained from the bias-corrected climate projection: (1) the bias-corrected rainfall

projections from the CMIP6 model output of the MRI-ESM2-0 model group and (2) the potential evapotranspiration derived from the bias-corrected near-surface air temperature projections from the CMIP6 model output of the MRI-ESM2-0 model group and the bias-corrected surface downwelling shortwave radiation and TOA incident shortwave radiation projections from the CMIP6 model output of the GFDL-ESM4 model group, based on two RCP scenarios (RCP4.5 and RCP8.5).

### 2.4.3 Groundwater flow model setup

The MODFLOW6 model, which solves the Darcy three-dimensional groundwater flow equation using the control-volume finite-difference (CVFD) method, is used to perform the groundwater flow simulation in this study. The model is built using the MODFLOW python package (Bakker et al., 2016).

The MODFLOW model parameterization in this study is based on the combination of literature reviews, fieldwork, and laboratory experiments. The model's subsurface vertical discretization is based upon collated borehole data (Rahiem, 2020), interpreted as a 3-layer model: the upper aquifer as the top layer, the thin interspersing aquitard as the middle layer, and the lower aquifer as the bottom layer. The hydraulic conductivities of the soil were measured by a combination of slug tests in the field, laboratory tests, and private company reports. They were then recalibrated by minimizing the difference between the simulated and the observed groundwater table (Rusli et al., 2023a). The $K_h$ of the upper and the lower aquifer are found in relatively similar ranges between 0.15 and 0.58 meter per day, as they are formed by a solitary geological formation. The $K_v$ ranges between $3.0 \times 10^{-4}$ and $6.0 \times 10^{-4}$ meter per day. The aquitard is ten times less permeable than the aquifer. The storage parameters are obtained from private company pumping test reports, and the river-related parameters were previously calibrated for steady-state conditions. The initial condition is defined by our previous simulation results (Rusli et al., 2023a). The well package, related to groundwater abstraction, is set according to the scenario described in Section 2.3.2. The full description of the groundwater flow model parameterization, including the model parameter recalibration, is also reported in our previous studies (Rusli et al., 2023a, b).

Similar to the hydrological simulation, we split the forcing data and the simulation period into two windows; the baseline period between 2005 and 2015, and the future projection period on short-term (up to 2030), mid-term (up to 2050), and long-term (up to 2100) future. Both periods are forced by the groundwater recharge simulated by the Wflow_sbm model, resulting in three different inputs as the drivers of the groundwater flow model: the baseline groundwater recharge, the future RCP4.5 groundwater recharge, and the future RCP8.5 groundwater recharge. With the combinations of different inputs and boundary conditions, six different outputs are produced from the groundwater flow simulation.

## 3    Results

### 3.1    Climate projection

The results for the climate projection are visualized by comparing the discussed climate variables - rainfall, temperature, solar radiation, or potential evapotranspiration - during the baseline period and the future period. Quantitatively, the variables

**Table 2.** Application of ISIMIP3b bias correction to the rainfall estimates projection from MRI-ESM2-0 model group (in millimeter)

| | 'Ground truth' | Historical data* | | RCP4.5 | | RCP8.5 | |
| --- | --- | --- | --- | --- | --- | --- | --- |
| | CHIRPS | Pre-** | Post-** | Pre-** | Post-** | Pre-** | Post-** |
| | (1) | (2) | (3) | (4) | (5) | (6) | (7) |
| Min | 0.00 | 0.00 | 0.00 | 0.00 | 0.00 | 0.00 | 0.00 |
| Lower quartile | 0.00 | 0.08 | 0.00 | 0.06 | 0.00 | 0.06 | 0.00 |
| Median | 4.67 | 0.97 | 4.17 | 2.81 | 5.45 | 2.39 | 5.18 |
| Mean | 7.61 | 5.74 | 7.62 | 6.87 | 8.70 | 6.64 | 8.52 |
| Upper quartile | 12.50 | 9.19 | 12.53 | 11.41 | 14.60 | 10.92 | 14.26 |
| Max | 111.66 | 91.45 | 111.66 | 88.13 | 131.67 | 121.70 | 123.90 |

*historical data of the climate projection products have the temporal coverage of the baseline period (1981 - 2015), and **the pre- and post- columns represent the values pre- and post- bias-adjustment in the future periods (2015 - 2100).

are compared using their statistical attributes: minimum and maximum values, mean, and lower, middle, and upper quartile. Visually, they are also represented in boxplots for easier interpretation, categorized by monthly estimates.

### 3.1.1  Bias corrected rainfall projection

The statistical attributes of the rainfall estimates pre- and post- bias correction, as well as ones of the future scenarios, are summarized in Table 2. In the MRI-ESM2-0 model group 'historical' output pre-bias correction, the rainfall is, surprisingly, projected to decrease in general compared to the CHIRPS estimates (column 1) in both RCP4.5 (column 4) and RCP8.5 (column 6). These values are, however, prior to bias correction. The same trend of climate model underestimation in almost every statistical distribution is also found during the baseline period (column 1 and column 2), therefore bias correction is 310 essential to be implemented. By applying the ISIMIP3b bias adjustment, we come up with baseline rainfall estimates that represent a better statistical fit to the CHIRPS estimates (column 3). Consequently, we apply the bias correction to the future scenario of RCP4.5 (columns 5) and RCP8.5 (columns 7).

To observe the seasonal impact on the projected climate scenario, we plot the monthly rainfall of the baseline period of CHIRPS and the future period of the bias-corrected RCP4.5 and RCP8.5 projections in Figure 4. The boxplots represent the 315 median and interquartile range of the interannual estimates, while the whiskers show the estimates' range. During the rainy season between October and March, we can see that the projected rainfall has an increasing trend, with higher monthly rainfall especially from November to January. A contrasting trend is shown in the dry season between April and September, with lower monthly rainfall especially from July to September. In short, the wet period is projected to become wetter, and the dry period is projected to become drier. We can also observe only small differences in the statistical quartiles between the two RCPs, with 320 similar widths, mostly, between the orange and red boxes. The width of the minimum and maximum values, however, is more apparent. The magnitudes of the hydrological extremes are projected to be more pronounced in the future, therefore floods and droughts are predicted to be more severe than they currently are.

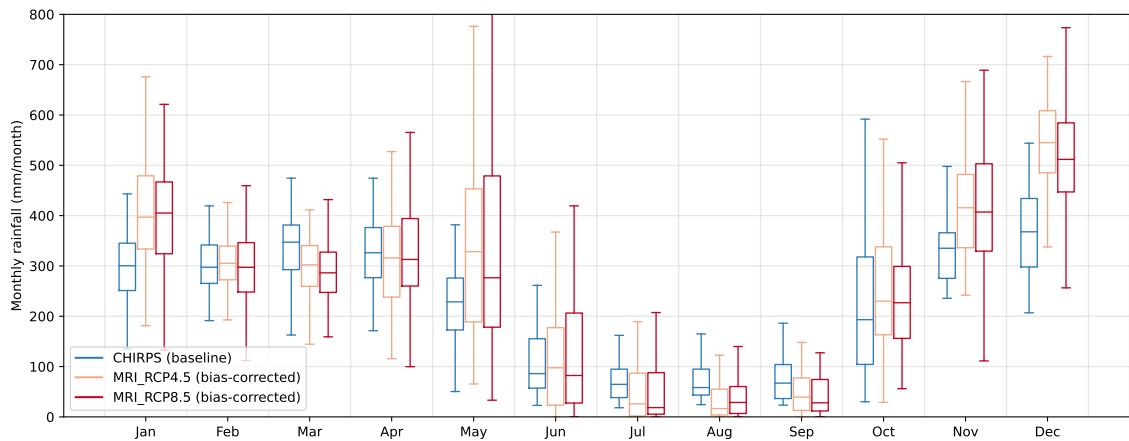

**Figure 4.** The comparison between the rainfall monthly statistics in the baseline period and the two future climate scenarios (RCP4.5 and RCP8.5). The future rainfall estimates are bias-corrected using the ISIMIP3b bias correction method. The box and whisker plots represent the median, interquartile range, and range of the interannual estimates.

### 3.1.2 Bias-corrected potential evapotranspiration projection

We apply the bias-correction method of ISIMIP3b to the near-surface air temperature, surface downwelling shortwave radiation, and top of atmosphere (TOA) incident shortwave radiation, in a similar fashion as one to the rainfall estimates. Figure 5a and 5b show the monthly near-surface air temperature and radiation projections, respectively. The monthly average temperature sharply increases from the baseline period of the aggregated ERA5-Land hourly estimates to the future period of the bias corrected CMIP6 projections in all statistical attributes; quartiles, average, interquartile range, and extreme values. The temperature, on the long-term average, is projected to be warmer by 2.21°C and 2.72°C in RCP4.5 and RCP8.5 scenarios, respectively. The projection on global radiation is based on the GFDL-ESM4 model group, with a tendency of slightly less surface downwelling shortwave radiation in the future. The TOA incident shortwave radiation remains almost constant throughout. On the radiation variable, the two different climatic scenarios of RCP4.5 and RCP8.5 do not seem to differ a lot in their statistical values.

We use the three variables above to calculate potential evapotranspiration using the method of de Bruin et al. (2016), with the result shown in Figure 5c. The range of the estimates is visually inconsistent between the baseline and the future period, as the future period is calculated in monthly time steps according to the temporal resolution of the radiation projections. Therefore, the variation between the statistical distribution is lower compared to ones of the baseline period, where it is available in the daily time step. The difference in the magnitude is considered low, as the highest difference in average daily potential evapotranspiration is less than 0.5 mm per day. Without looking at the seasonal variance, the annual daily potential evapotranspiration averages of the baseline period, the future RCP4.5 scenario, and the future RCP8.5 scenario are 3.24, 3.26, and 3.23 mm per day, respectively, which are relative to insignificant differences.

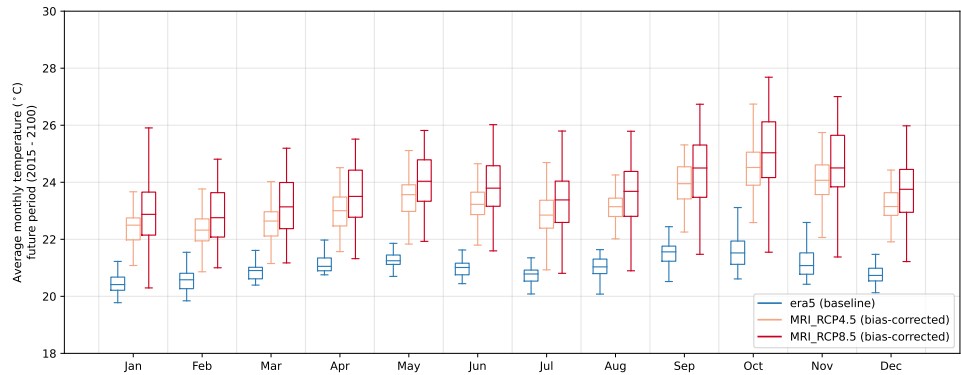

(a) Comparison between the temperature in the baseline period and the two future climate scenarios (RCP4.5 and RCP8.5).

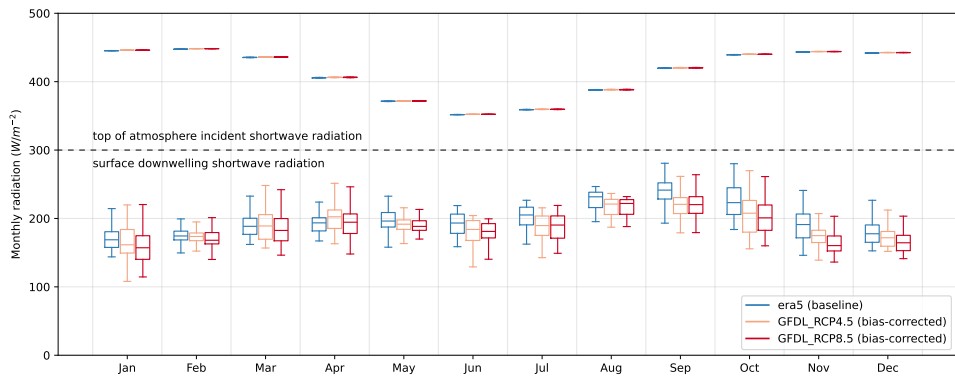

(b) Comparison between the radiation in the baseline period and the two future climate scenarios (RCP4.5 and RCP8.5).

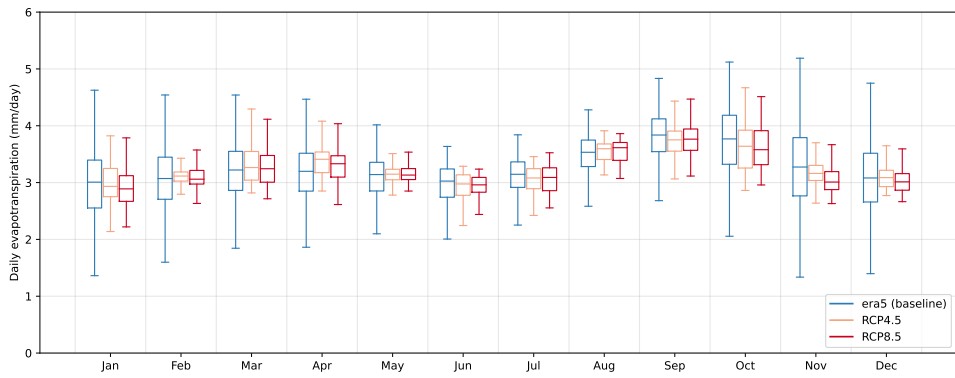

(c) Comparison between the potential evapotranspiration in the baseline period and the two future climate scenarios (RCP4.5 and RCP8.5).

**Figure 5.** The results on bias-corrected projections on three variables: (a) temperature, (b) radiation, and (c) potential evapotranspiration. The box and whisker plots represent the median, interquartile range, and range of the interannual estimates.

## 3.2 Groundwater recharge projection

The projected rainfall and potential evapotranspiration are used to force the Wflow_sbm model, resulting in projected ground-water recharge. As the change in groundwater recharge in mm per day unit is relatively small, we accumulate the daily recharge to monthly recharge to produce more intuitive figures. The seasonal pattern of the monthly groundwater recharge is shown in Figure 6a. The values intrinsically represent the number of days in each month, therefore the groundwater recharge in the non-31-day months is lower than those in the 31-day months. The results are confirmed to be consistent in all the short-term, mid-term, and long-term future assessments of groundwater recharge, with consistent and slightly increasing groundwater recharge during wet and dry seasons, respectively.

It can be seen that after the start of the wet season when the soil moisture starts to be saturated (December) to the beginning of the dry season when the soil's maximum capacity for storing water is still attained (May), the groundwater recharge could no longer increase despite the increasing rainfall projection during the wet season. On the other hand, after the start of the dry season when the soil moisture starts to dry up (June) to the beginning of the wet season when the void spaces in the subsurface are still available for water to fill into (November), the magnitude of the groundwater recharge is relatively more subject to change. Having said that, during the latter period, there is only a small difference in the median and extreme values of the groundwater recharge, although the quartile values vary. Based on the median values, the largest difference in groundwater recharge is projected to occur in either September (for RCP4.5) or October (for RCP8.5), in an order of magnitude of a minuscule increase of 1.27% and 1.79%. In all other months except June, the groundwater recharge is projected to either, despite being a very small change, increase, or remain relatively constant. Annually, the average groundwater recharge during the baseline period of 315.1 mm per year is projected to insignificantly increase to 316.1 mm per year for the RCP4.5 and 316.4 mm per year for the RCP8.5 scenario. On average, the absolute relative change in the rainfall, temperature, and groundwater recharge estimates under the RCP4.5 scenario, respectively, are 31.03%, 10.71%, and 0.36%. The similar changes for the RCP8.5 scenario are 28.36%, 13.10%, and 0.44%. Figure 6b displays the comparison of the magnitude of change in the rainfall and potential evapotranspiration on the left axis as well as the groundwater recharge on the right axis with a ten times larger scale, all in monthly average values. The average value is chosen over the median value, specifically for this figure, to take into account the projected extreme values. From the results, the change in the groundwater recharge is found to be far less responsive compared to its driver, especially the rainfall.

The results could indicate two possibilities. First, it reveals the dominant role of the already-saturated soil in controlling the groundwater recharge processes in the Bandung groundwater basin. When the soil's maximum capacity is already reached, more rainfall does not directly affect the recharge processes. Instead, it increases surface runoff, river discharge, and more water loss to evapotranspiration; the latter occurs only when the PET allows. However, as shown in Figure 5c, the projected change of PET varies between months. During the wet season, in particular, the PET is even projected to slightly decrease. This concept is agreed upon by the hydrological model simulation under the projected climatic scenarios. Figure 7a and 7b, respectively, show the boxplot of the simulated river discharge and actual evaporation in the baseline and future periods. These graphs indicate an increase in the river discharge variable and a slight decrease in the actual evaporation. Specifically for the river

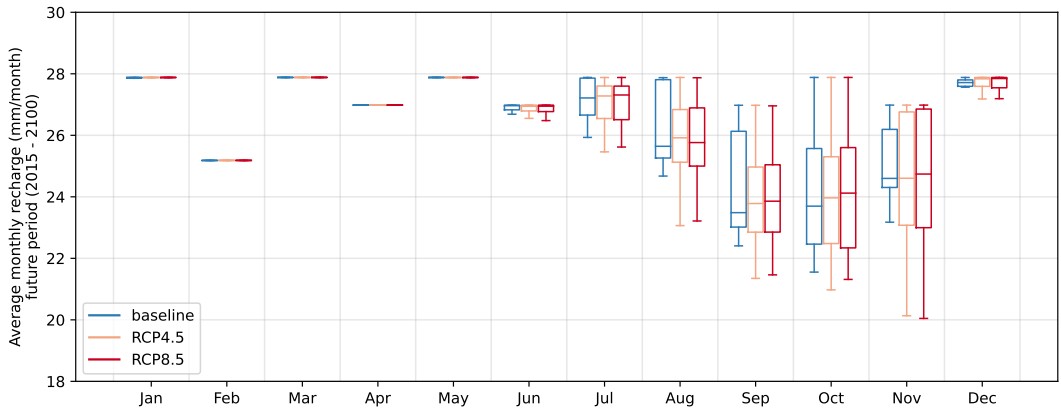

(a) Comparison between the simulated groundwater recharge in the baseline period and in the two future climate scenarios.

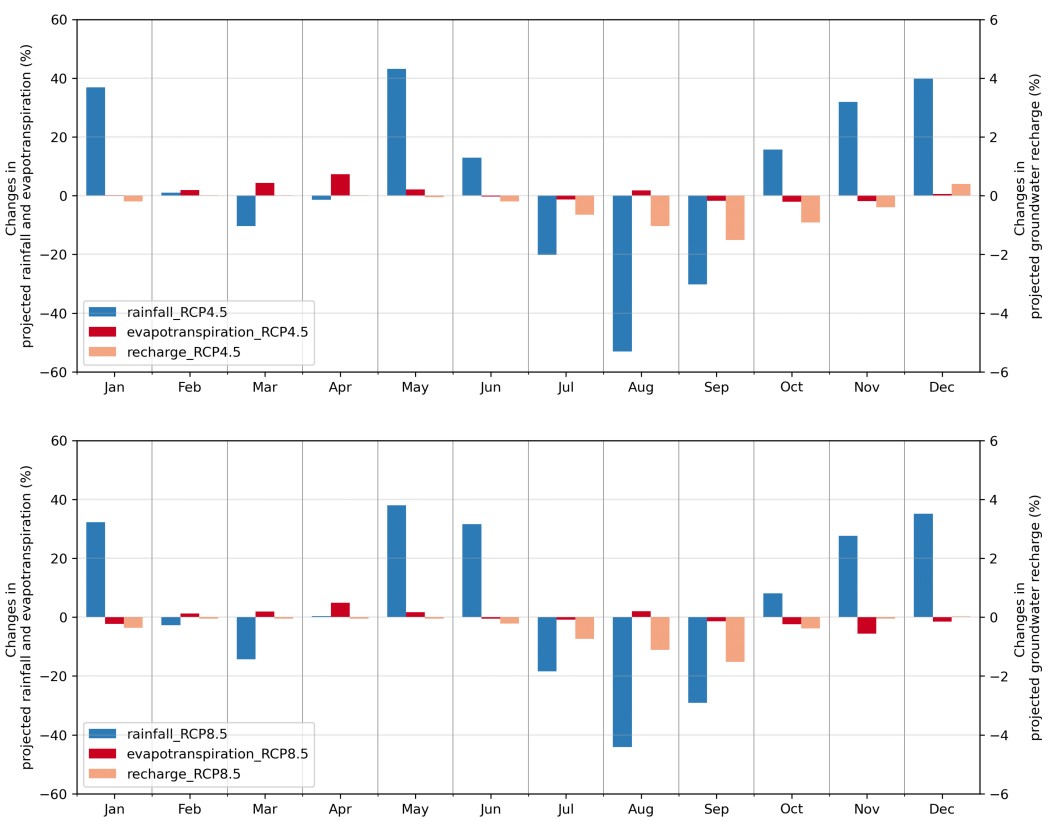

(b) Comparison between the projected average monthly rainfall, potential evapotranspiration, and groundwater recharge.

**Figure 6.** The results of (a) groundwater recharge projection and (b) its value relative to its driver of rainfall and potential evapotranspiration. The values are averaged over the future period, annually, up to 2100.

discharge, the projected amplification is remarkably apparent, and the magnitude of the rise is larger the further the projections are, following the trend shown in the rainfall projection. The results on actual evaporation follow the ones of the projected PET, with largely a minor decrease in values. These changes balance out the increment of the rainfall forcing, supporting the outcome of groundwater recharge that shows relatively steady consistencies despite the increase in the hydrological forcing dataset. In a water balance term, the amplified magnitude of inflow in the future is followed by the rising outflow, causing the changes in the other outflux of groundwater recharge to be minimal. Secondly, the impact of changing climate variables on groundwater recharge could be hindered by the model coupling limitation, causing the model to underestimate the degree of influence. The fact that the hydrological and the groundwater flow model are one-way coupled, instead of fully coupled, caps the two-way feedback nature of groundwater recharge. Physically, groundwater recharge is controlled by both the surface processes (represented by the hydrological simulation in this study) and the amount of space available to be recharged (represented by the groundwater flow model). However, in our one-way model coupling scheme, groundwater recharge is fully determined by the surface processes. These two notions are further discussed in Section 4.2.

### 3.3 Groundwater level projection

The combination of two climatic and three groundwater abstraction scenarios results in six outcomes. For each outcome, there are three temporal checkpoints assigned as the milestone of the assessment: the short-term, the mid-term, and the long-term future in 2030, 2050, and 2100, respectively. There are also two layers of aquifers, the unconfined and the confined aquifers, to be assessed. As there are a lot of numbers to unpack, we discuss the results as per the abstraction scenario.

Generally, we focus on the maximum drawdown values of the groundwater table/piezometric head, as the change in the groundwater head is found to be highly localized, both from the simulation and observation perspective. Furthermore, the baseline groundwater abstraction area is estimated 'only' at 27.3% of the total basin area for the domestic groundwater abstraction, and even at 4.7% for the more intensive industrial abstraction. Taking the average or median value for the whole groundwater basin, therefore, would not be suitable to represent the severity of the groundwater abstraction impact, considering the high number of cells involved in the numerical simulation.

Under the increasing groundwater abstraction scenario, the groundwater level is projected to continue decreasing. The numbers are worse for the confined aquifer, as the groundwater abstraction is spatially more concentrated with higher abstraction rates, as well as having a much smaller storage coefficient compared to one in the unconfined layer. Under the RCP4.5 scenario, the maximum piezometric head drawdown for the confined aquifer is projected to be at 10.04 m in 2030, 19.98 m in 2050, and 48.79 m in 2100. Under the RCP8.5 scenario, the numbers are also concerning for the unconfined aquifers, as the groundwater table is projected to dwindle to up to 3.38 m and 3.40 m in the long run under the RCP4.5 and RCP8.5 scenarios, respectively. Based on the drawdown area, the trend shows that the impacted area is enlarged as the groundwater abstraction area expands. In the unconfined aquifer, respectively 74.6%, 80.5%, and 87.2% of the groundwater basin area is projected to experience groundwater table drawdown in 2030, 2050, and 2100 under the RCP4.5 scenario, obviously with varying magnitude. For the confined aquifers, the numbers representing the depression area are 70.5%, 72.9%, and 75.4%. Under the RCP8.5 scenario, these numbers are pretty much consistent, with a maximum of less than 1% difference from the RCP4.5 results.

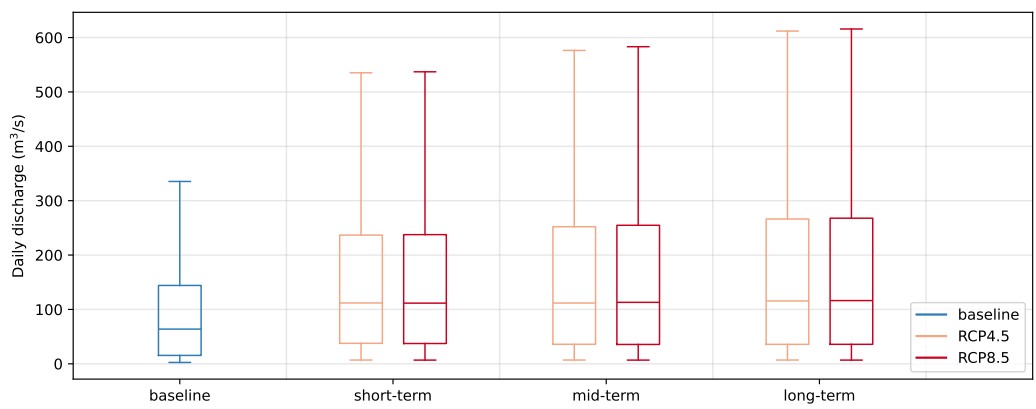

(a) Baseline and future periods' boxplot of simulated river discharge.

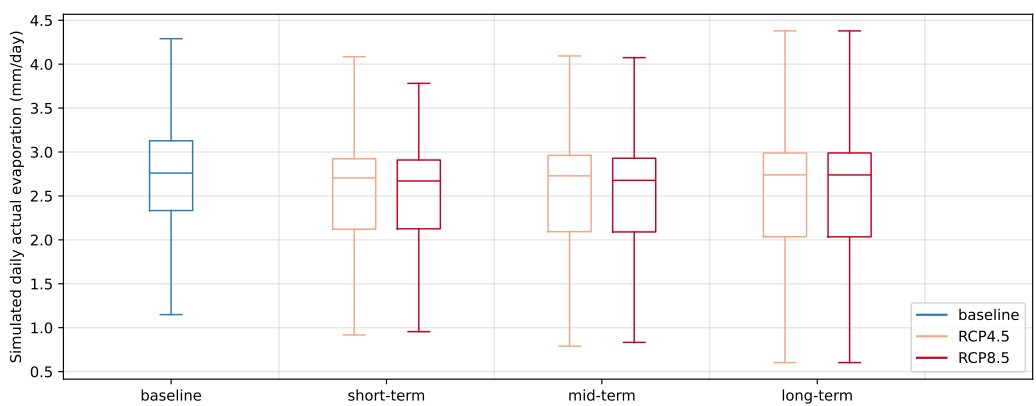

(b) Baseline and future periods' boxplot of simulated actual evaporation.

**Figure 7.** The boxplot visualization of the simulated projected (a) river discharge and (b) actual evaporation.

The second abstraction scenario portrays the projection of the current situation in the study area. Should the anthropogenic pressure remain the same, the groundwater level, as expected, is projected to remain dwindling. The maximum confined piezometric head drawdown for 2030, 2050, and 2100 under the RCP4.5 scenario are 7.14, 15.25, and 29.51 m. The numbers for the RCP8.5 scenario are very similar: 7.14, 15.28, and 29.51 m. Similar to the increasing abstraction scenario, the unconfined aquifer groundwater table drawdown is noticeably lower compared to that of the confined aquifer, with a maximum drawdown of 2.58 and 2.60 m in 2100 for RCP4.5 and RCP8.5, respectively. The drawdown area increases, with up to 84.7% and 75.0% of the groundwater basin area experiencing long-term dwindling groundwater heads in the unconfined and confined aquifer. Despite the lesser extent of the impacted area relative to the previous scenario, the area of decreasing groundwater head remains relatively dominant in comparison to the basin's total area. To visualize the time-series increasing impact of the groundwater head drawdown, Figure 8 shows the propagation of the groundwater head decline in the Bandung groundwater basin under the RCP4.5 climatic scenario and the constant groundwater abstraction anthropogenic scenario.

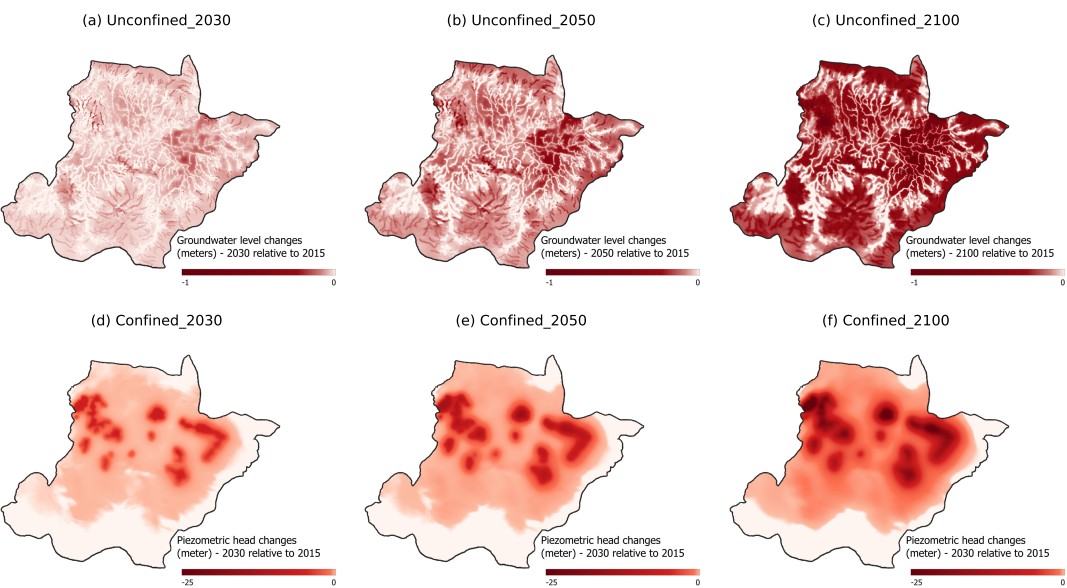

**Figure 8.** The spatial distribution of the projected groundwater head change under the RCP4.5 and the constant groundwater abstraction scenario. Figures (a), (b), and (c) represent the unconfined aquifers in the short-term, mid-term, and long-term future, respectively. Meanwhile, Figures (d), (e), and (f) represent the confined aquifers in the short-term, mid-term, and long-term future, respectively.

The third abstraction scenario projects the groundwater abstraction to decrease, influenced by the relocation of Indonesia's capital city. Under this scenario, the groundwater is simulated to be partially replenished between 2050 and 2100. This is indicated by the projected piezometric head drawdown that reaches 12.61 m in 2050, but is calculated at 11.75 m in 2100. Granted that 11.75 m remains a net negative of groundwater head in the future, however, subsurface flow (and therefore replenishment) requires a long time to reach an equilibrium. The fact that the piezometric head drawdown decreases signals an improving situation under scenario three. Having said that, the groundwater table in the unconfined aquifer is projected to decrease up to 2.58 and 2.93 m under the RCP4.5 and RCP8.5 scenarios, respectively. However, its drawdown area is projected to be smaller, going from 80.8% and 83.6% in 2030 for RCP4.5 and RCP8.5 respectively to 75.5% and 76.7% in 2100. The response of groundwater replenishment is unique between the aquifer layers; the unconfined aquifer primarily reduces the drawdown area, while the confined aquifer relaxes the magnitude of the piezometric head drawdown. Nevertheless, it presents an opportunity for groundwater replenishment in the future given the right policy and management in the study area.

Table 3 summarizes the maximum groundwater head drawdown in both the climatic and anthropogenic scenarios, checkpoints (short-term, mid-term, and long-term future), and layers (unconfined and confined aquifers) in focus. The table shows that the anthropogenic factor holds a more dominant influence on future groundwater storage relative to the climatic factor. This is indicated as different RCPs in an abstraction scenario offer smaller changes in values compared to different abstraction scenarios under a climatic scenario. We can also see the increase of the impact over time and the more severe impact on the confined groundwater head compared to the unconfined groundwater table.

**Table 3.** Summary of projected maximum groundwater head drawdown under multiple climatic and anthropogenic scenarios (in meter)

| | | Increasing abstraction | | Constant abstraction | | Decreasing abstraction | |
|---|---|---|---|---|---|---|---|
| | | RCP4.5 | RCP8.5 | RCP4.5 | RCP8.5 | RCP4.5 | RCP8.5 |
| | | (1) | (2) | (3) | (4) | (5) | (6) |
| 2030 | Unconfined | 0.98 | 0.97 | 0.89 | 0.89 | 0.89 | 0.97 |
| | Confined | 10.04 | 10.04 | 7.14 | 7.14 | 7.11 | 7.11 |
| 2050 | Unconfined | 1.69 | 1.69 | 1.49 | 1.49 | 1.49 | 1.69 |
| | Confined | 19.98 | 19.98 | 15.25 | 15.28 | 12.60 | 12.61 |
| 2100 | Unconfined | 3.38 | 3.40 | 2.58 | 2.60 | 2.58 | 2.93 |
| | Confined | 48.79 | 48.79 | 29.51 | 29.51 | 11.55 | 11.75 |

## 3.4 Groundwater storage projection

As stated above and shown in Figure 8, the drawdown due to the groundwater abstraction is highly congregated spatially. In the unconfined layer, the drawdown area is distributed under the abstraction area, while the area close to water bodies is less impacted. This occurs due to surface water and groundwater interaction, where the groundwater table is also regulated by the surface water elevation aside from the subsurface flow. In the confined aquifer, the highly elevated area on the surface is much less impacted compared to the one in the overlying aquifer, and the drawdown area is, in general, directly located under the abstraction area. Using only the groundwater head, although useful, could not capture the whole picture of the basin's groundwater regime projection. Therefore, we also assess the groundwater projections from the perspective of the integrated aquifer water balance and the integrated cumulative storage changes over time.

Figure 9 shows the magnitude of projected recharge relative to the projected groundwater abstraction. The groundwater recharge, as suggested by the above simulations, is relatively constant, with small fluctuations coming from the seasonal temporal variability. The less dominant effect of climatic variables is also visible from the similarity of groundwater recharge values between the two RCPs. On the other hand, the groundwater abstraction before 2020 was similar among the scenarios. After the baseline period, it starts to diverge from the increasing to decreasing projections. The figure also shows that between 2020 and 2025, the volume of groundwater abstraction starts to surpass that of groundwater recharge, exerting the turning point in the aquifer water balance components. It appears that pre-2023, the total integrated volume of the groundwater abstraction is still slightly below the total integrated volume of the groundwater recharge. It does not mean that the whole basin water table is rising, though, as spatially, both the displayed variables are not uniformly distributed. Added to that is the fact that groundwater moves relatively slowly compared to surface water, therefore it requires a long time for the groundwater in the mountainous region to flow downstream, compensating the groundwater abstraction in the urban area. Past 2023, under the scenario where the groundwater abstraction increases, the total volume of groundwater abstraction is projected to surpass the groundwater recharge, making it implausible to replenish the groundwater storage. In the other two groundwater abstraction scenarios, the total groundwater abstraction volume is projected to be still slightly below the groundwater recharge. However, it does not

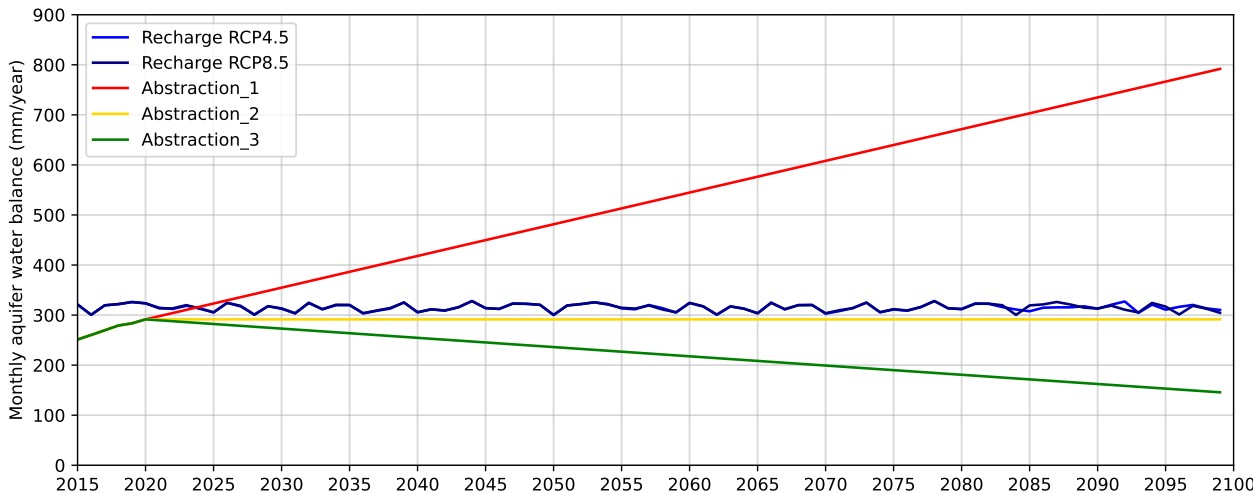

**Figure 9.** The aquifer water balance of groundwater recharge and groundwater abstraction from 2015 to 2100 for all climatic and anthropogenic scenarios.

mean that the groundwater storage is replenished. As mentioned above, groundwater flows at a low velocity, therefore reaching an equilibrium over the integrated basin area would require a long time. Secondly, the groundwater storage is also controlled by other fluxes, and most importantly the surface water and groundwater interaction. The fact that the baseflow of the river (partially indicated in Figure 7a) does not project a decrease in values points out that the groundwater constantly supplies the
river baseflow, therefore decreasing the groundwater storage.

     Similar to the projected groundwater level assessment, six outcomes are produced from the combination of two climatic and three anthropogenic scenarios. Figure 10 shows the trajectory of each outcome in terms of its accumulated groundwater storage changes relative to the one in 2015 as the benchmark. Consistent with the aforementioned results, the difference between climatic scenarios is very thin, as lines with the same color almost intersect. However, the impact of the diverging
abstraction scenario is visually apparent and even results in diverging groundwater storage change. Using the gradient of the storage depletion accumulation for scenario two as the benchmark, it gets up to 3.43 times steeper for scenario one and up to 0.40 times milder for scenario three during the most extreme year, both propagating in curve-shaped lines. The lines' shapes deliver important messages, as they indicate the uniquely diverging results of deteriorating, sustained, and improving groundwater storage depletion under the first scenario, second, and third scenarios, respectively. It is also notable that despite
the indication of confined groundwater replenishment from the groundwater level perspective, assessment of the groundwater storage change suggests otherwise, further discussed in Section 4.2.

     Under the constant abstraction scenario, the Bandung groundwater storage is projected to lose almost $2 \times 10^{10}$ m$^3$ in the upcoming 85 years. While the number might seem exaggerated, it is actually equivalent to an average of 0.54 mm per day of storage depletion. The rate of storage depletion is relatively constant throughout the short-term, mid-term, and long-term
future in this scenario, as per the abstraction rate. Under the increasing abstraction scenario, the long-term storage depletion

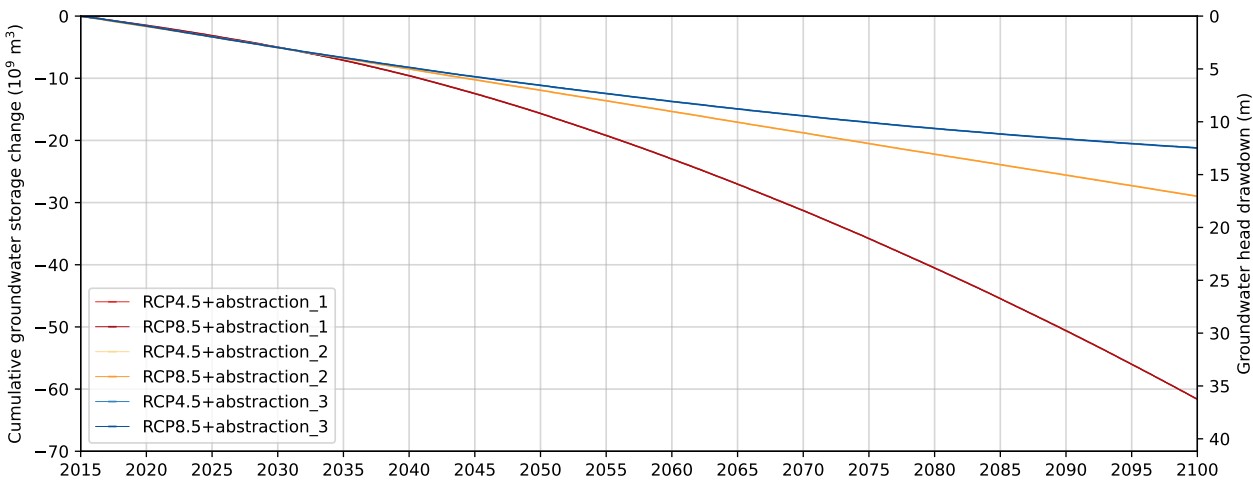

**Figure 10.** The cumulative projected groundwater storage changes from 2015 to 2100 for all climatic and anthropogenic scenarios.

is projected at 1.17 mm per day on average. This is a result of an escalating depletion, as the storage loss is averaged at 0.53 mm per day between 2015 and 2030, 0.86 mm per day between 2030 and 2050, and 1.48 mm per day between 2050 and 2100. Under the decreasing abstraction scenario, the long-term storage depletion is projected at 0.40 mm per day on average. This is the result of a withering depletion, as the storage loss is averaged at 0.54 mm per day between 2015 and 2030, 0.49 mm per day between 2030 and 2050, and 0.33 mm per day between 2050 and 2100.

## 4 Discussion

### 4.1 Projections uncertainty

In climate projection-related studies, uncertainties are unavoidable as they are propagated from multiple sources: the natural climate variability, the climate model, and the emission scenario (Latif, 2011). The natural climate variability is highly uncertain as Buser et al. (2010) suggested the extrapolative nature of climate variables which involved different biases in the scenario and the control period. Each model also has different responses to climatology and perturbation components uncertainties, for example, as stated by Adachi et al. (2019). To tackle the wide range of uncertainty bounds, many studies propose using an ensemble of climate projection products (Hawkins et al., 2016; Rajczak and Schär, 2017). Meanwhile, other studies promote the efficiency of bias correction to reduce the uncertainties of climate projection products as such a method takes into account the 'ground-truth' estimates of the corresponding climate variables (Lange, 2019; Wu et al., 2022).

The climate projections resulting from different model groups have varying estimates and uncertainties. However, we believe that these uncertainties have been addressed in our study. First, the focus of the hydrological simulation is groundwater recharge, which is not solely influenced by climate variables. Indeed, climate variables play an important role, as they provide the input to the basin that further generates recharge. However, the control on effective groundwater recharge is also regulated

by catchment features, such as soil moisture capacity, surface water cycle, and subsurface properties. With the other simulation components' relatively known values, in this case, the Wflow_sbm model parameterization, the groundwater recharge estimates, therefore, are not singlehandedly impacted by the uncertainties of the climate projections. Second, the primary goal of the study is to project the groundwater availability. With a relatively lower range of projected change in future groundwater recharge, previous studies suggested the anthropogenic factor to play a larger role and involve a wider range of future uncertainties compared to the climatic factor (Mustafa et al., 2019; Aslam et al., 2022). In our scenario, the three abstraction scenarios that span from increasing to decreasing projection are expected to be able to outweigh the uncertainty bounds propagated from the climatic scenario. All the results from our projection in this study support the claim: in every scenario, the change in groundwater abstraction always produces significantly higher influences on the groundwater table and storage compared to the climatic scenario. Third, we also apply a bias correction method to the climate projection products, which has been proven in

previous studies to effectively reduce climate projection uncertainties (Rahimi et al., 2021; Wu et al., 2022). Table 2 columns (1), (2), and (3) show a remarkable improvement in the bias-corrected climate model output. By bias-correcting the projected climate variables and taking into account the historical high-resolution 'ground truth' data as the benchmark, we believe the uncertainties of the climate projection have been significantly reduced.

## 4.2 Impact assessment on future groundwater level projections

As shown in the simulation workflow (Figure 2), future groundwater status is projected by altering climate forcing input and groundwater abstraction as the boundary condition. The former imparts its contribution to groundwater recharge estimates. However, as shown in Figure 6b, there are significant differences in the impact of climate variables' changes on the surface and the subsurface component of the water cycle. The rainfall median, not considering the seasonal fluctuation, is projected to change (either increase or decrease) up to 31.03% and 28.36% for RCP4.5 and RCP8.5 respectively. On the other hand, the

projected change in the groundwater recharge is less than 1%, indicating a slower response of subsurface components to the climate change projection.

This result is limited, however, by the nature of the model one-way coupling. In our model setup, groundwater recharge is fully controlled by the surface processes and the pseudo-water table. Physically, groundwater storage depletion due to groundwater abstraction might dwindle the water table. By that process, there would be more space for water to infiltrate, indicating

two-way feedback between groundwater abstraction and groundwater recharge. A one-way coupled model, unfortunately, is not capable of incorporating such two-way processes into its simulation. This further proves that groundwater abstraction and basin properties possess equal, if not more, importance compared to the climate forcing in groundwater recharge projection. This looks site-specific, however, depending on the basin features, especially the land use/land cover type, the soil maximum capacity, and the subsurface properties. In regions with higher margins between the groundwater recharge and soil capacity,

meaning the soil condition is not generally wet, changes in climatic factors (i.e. effective precipitation) would have a higher influence on the changes in groundwater recharge. This also highlights the importance of basin-scale information in climate projection studies (Bhave et al., 2013; Jackson et al., 2015; Marcos-Garcia et al., 2023), which have been conducted largely in global scale studies.

Additionally, the projected groundwater recharge is also simulated under the assumption of constant soil characteristics represented by the $MaxLeakage$ parameter in the Wflow_sbm model. Meanwhile, soil characteristics such as soil permeability, rock formation, soil infiltration capacity, as well as soil type and structure, might evolve (Zhang and Wang, 2023), albeit on a much longer time scale than the surface features. In our opinion, changes in soil characteristics would influence the $MaxLeakage$ parameter, therefore impacting the simulated projected groundwater recharge. While these changes are gradual, they are likely to affect recharge over time, potentially causing significant deviations from current projections. However, the tendency of how soil characteristics evolve long term, particularly regarding groundwater recharge generation, remains uncertain whether it will increase (Cook et al., 2022) or decrease (Wu et al., 2020a). Whereas, accurate future projections require constant soil monitoring and modeling. This further highlights the importance of data, especially soil-related data and hydrological information as the benchmark for model calibration and verification. Consistent data assimilation that is validated through updated observation and simulation would decrease the uncertainties, making it possible to make the $MaxLeakage$ parameter setup dynamic over time.

The uncertainties of future anthropogenic factors, considering their large influence, should be the primary focus in future groundwater management. Figure 9 emphasizes the importance of the current policies in managing groundwater abstraction, as between the years 2020 and 2025, the annual groundwater abstraction is estimated to surpass the total annual recharge volumetric-wise. This also shows that the previous depleting groundwater storage is controlled by the fluxes within the groundwater storage water balance components, mainly the surface-groundwater interaction. This is supported in Figure 8, where the groundwater table in the unconfined storage depletes mostly in the mountainous region, contributing to conserving the river baseflow due to the groundwater abstraction along the river downstream part. While the climatic factor is relatively intractable, groundwater abstraction activities, in terms of rates, volumes, and spatial distribution, are relatively manageable through regional/local groundwater policies, not to mention its more influential impact on the groundwater regime. A previous study suggests that groundwater abstraction even holds a higher influence on river baseflow compared to changes in climate (Taie Semiromi and Koch, 2020). Improving the understanding of subsurface response and bridging the key gap between science and policy on the matter of groundwater abstraction should be the main focus and responsibility of all involved stakeholders.

The simulation results also reveal the importance of multi-perspective assessment in groundwater regimes. On one side, Table 3 implies that the groundwater situation in the confined aquifer is improving under scenario three of groundwater abstraction, where the maximum piezometric head drawdown in 2100 is lower than one from 2050. On the other side, Figure 10 shows that the groundwater storage is still depleting, shown by the negative gradient of all the lines, including ones from the abstraction scenario three. Such discrepancies occur as the two assessment variables, the groundwater head and the groundwater storage, represent two different dimensions of the groundwater status. The groundwater head represents a point, or a single grid, value that constitutes local features, while the groundwater storage evaluates the basin-integrated response. Referring to only one assessment variable could lead to a misunderstanding of the process of the groundwater flow system. We discuss the interpretation of the two conflicting numbers in the following section.

### 4.3    Opportunities for groundwater replenishment

As shown in Figure 10, the different gradients represent diverging directions of the groundwater storage over time. While all the results accumulate negative changes, the rate at which the groundwater storage is depleting differs among scenarios. With a decreasing groundwater abstraction scenario in the future, the depletion rate is projected to decline, as expected. Admittedly, groundwater replenishment might take a comparably long time to reach a 'new equilibrium', considering the subsurface low flow velocity. The current declining, but slower, groundwater storage depletion, therefore, could be interpreted in two ways, either (1) the groundwater storage is indeed still in a deterioration trend, or (2) the groundwater storage is actually being replenished, but has not yet reached the 'new equilibrium' state as the subsurface time scale is longer than the surface's. The latter hypothesis is supported by the values in Table 3, which at first glance might seem inconsistent with the results in Figure 10. By any means, the results in scenario three, which is highly possible due to the capital city relocation plan, suggest an opportunity for future groundwater replenishment, although it takes some time to yield a positive turning point. Consistent future groundwater head monitoring in the study area could provide crucial insight, which will assist in deriving adaptation policies in response to the capital city relocation.

We also notice the different responses of groundwater 'replenishment', which is contrasting between the aquifer layers. In the unconfined aquifer, the primary response of the groundwater improvement is to reduce the impacted drawdown area. While the groundwater table is still dwindling in all future checkpoints, there are smaller (simulated) drawdown areas in 2100, even compared to ones in 2030. In contrast, the confined aquifer relaxes the magnitude of the piezometric head drawdown while maintaining the drawdown area. This, presumably, is directly related to the spatial distribution of groundwater abstraction. The groundwater abstraction applied in the unconfined aquifer is more widespread with lower rates of abstraction. Therefore, the drawdown area is highly dependent on the spatial distribution. In the opposite to the unconfined aquifer, the groundwater abstraction applied in the confined aquifer is more concentrated with intense rates of abstraction. Decreasing the rate, consequently, delivers noticeable influences on the stressed piezometric head. This reveals an important opportunity for future groundwater policies: the governance of groundwater abstraction authorization should include not only the abstraction rate limitation but also the consideration of future and integrated geospatial planning of the study area.

## 5    Conclusions

In this study, we develop groundwater status projection under multiple climatic and diverging anthropogenic scenarios. We simulate the groundwater recharge projection using the hydrological model Wflow_sbm. The climate projection forcing is taken from the CMIP6 MRI-ESM2-0 model group, including the projected rainfall as well as the projected temperature and radiation data to estimate potential evapotranspiration. We force the Wflow_sbm model with two RCP scenarios: RCP4.5 and RCP8.5. Further, using the groundwater recharge projection as the groundwater flow driver, we simulate the subsurface flow under groundwater abstraction as the boundary condition. We develop three diverging groundwater abstraction scenarios; increasing groundwater abstraction as the most common approach, constant abstraction as the benchmark, and decreasing abstraction as a possibility. We take the Bandung groundwater basin in Indonesia as our test case, located nearby Jakarta,

the current capital city of Indonesia. The Bandung groundwater basin has a wide range of uncertainties in terms of future groundwater abstraction, in response to the Indonesia capital city relocation plan, therefore nicely covering the three developed anthropogenic scenarios.

We applied the bias correction method of ISIMIP3b to the CMIP6 climate projection data before forcing it to the Wflow_sbm model. The bias correction reduces the uncertainty of the climate variables' projection, as the bias-corrected historical data show consistent statistical distributions to the 'ground-truth' data. Future rainfall and temperature median are projected to change by 31.03% and 10.71% under RCP4.5, and 28.36% and 13.10% under RCP8.5. Future groundwater recharge projection reveals the dominant control of the soil component in generating the groundwater recharge in the study area. The fact that there is less than a 1% change projected for the groundwater recharge variable under both climatic scenarios shows that most of the time, the recharge is already at its maximum capacity. During the rainy season, rainfall intensification could not generate more recharge. On the other hand, during the dry season, increasing rainfall drives higher recharge, however, during the dry period that is projected to be even drier, the deficit of groundwater recharge almost balances out those additional recharge.

As expected, under the increasing and constant groundwater abstraction scenario, the groundwater status is projected to drop. The maximum groundwater head drawdown increases over time, the drawdown area expands, and the groundwater storage depletes. However, a positive sign of groundwater replenishment potential is shown under the decreasing groundwater abstraction scenario, despite the conflicting numbers shown between the point-based groundwater level assessment and the basin-integrated groundwater storage assessment. Despite being slow and occurring between 2050 and 2100, there is a sign that the groundwater storage is going in the right direction to be refilled. It is also indicated that the groundwater table is highly regulated by not only the volume and rate but also the spatial distribution of groundwater abstraction.

Comparing the results of climatic and anthropogenic impact, we conclude that groundwater abstraction propagates a more significant impact on the simulated future groundwater availability in areas with high groundwater dependency, such as the Bandung groundwater basin. Yet, this has to be approached with a case-by-case perspective considering the spatial variability of the climatic factor, anthropogenic factor, and hydrology and hydrogeological properties of the area in focus. The context of the projection of rainfall, evaporation, and river discharge is also important in applying such analysis to a wider context beyond a case study. The findings of this study are expected to assist in deriving and improving the current and future groundwater policies and management strategies, in particular for the Bandung groundwater basin, and in general for other groundwater basins around the world that face similar issues.

*Data availability.* The climate model data are publicly available on the Copernicus Climate Data Store website, which can be accessed online at https://cds.climate.copernicus.eu (last access: 31 July 2023). Other data used in this study are stored in the 4TU Research Data repository, which is available online at: 10.4121/d9706a2a-b77b-412f-a3aa-6e22bd8ddf4a

630 *Author contributions.* SR, VB, and AW, altogether designed the study conceptualization. SM provided the theoretical background and related literature. SR managed the input data for the one-way coupled model. AW prepared the model setup for the hydrological simulation, and VB supervised the development of the groundwater flow model. SR ran the model simulations, while VB, SM, and AW validated the results. SR wrote the original draft, and VB, SM, and AW contributed to reviewing the manuscript before its finalization.

*Competing interests.* One of the authors, Albrecht H Weerts, is a HESS editor.

*Acknowledgements.* The first author would like to acknowledge the Indonesia Endowment Fund for Education (LPDP) under the Ministry of Finance, Republic of Indonesia, for its scholarship funding support. This research is also significantly supported by the Office of Energy and Mineral Resources (ESDM) of the West Java Province by providing access to the available data. We acknowledge the World Climate Research Programme, which, through its Working Group on Coupled Modelling, coordinated and promoted CMIP6. We thank the climate modeling groups for producing and making available their model output, the Earth System Grid Federation (ESGF) for archiving the data 640 and providing access, and the multiple funding agencies that support CMIP6 and ESGF.

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
