# Peer review of "The impact of future changes in climate variables and groundwater abstraction on basin-scale groundwater availability"

_Hydrology and Earth System Sciences, 2024_

## Author Response (AR1)

Rebuttal documents for the manuscript entitled:

"The impact of future climate projections and anthropogenic activities on basin-scale groundwater availability", submitted to the Hydrology and Earth System Sciences Journal

Editor comments:

Based on the reviews, I received I recommend the paper to be reconsidered for publication after major revisions. Both reviewers have noted the scientific significance of the paper as "fair" as they have also mentioned that the paper targets a regional study and the methodology has already been published in previous studies. I recommend the authors revise their manuscript to highlight the novelty of their paper and its scientific significance and how the application of that methodology to a particular region can be leveraged.

Response to Referee #2:

We sincerely thank you for the effort you have allocated to manage the reviewing and open discussion process of our manuscript. Indeed, the comments on the scope and methods of the manuscript were mentioned by the referees. We have accommodated both issues in our latest revision.

We very much welcome your suggestion to revise the manuscript in highlighting the novelty of the paper. In the marked-up changed manuscript, Line 47 – 55, we have added almost a whole new paragraph revisiting our approach in interpreting the aim of our study. We acknowledge the first iteration of our manuscript was too biased towards one school of thought regarding climatic and anthropogenic influence on groundwater recharge, thus giving the impression of a study 'confirming' that one side of knowledge. In our new version, however, we address both opinions, and acknowledge the spatial variability of the two mentioned factors. While anthropogenic factors could be the dominant factor in regulating groundwater availability in some areas, climatic ones could also be highly influential in some other places. This depends on several variables, such as climatic condition, catchment/regional features and properties, groundwater abstraction, and so on. Thus, the novelty of this study is aimed to unravel the impact of both factors in such a groundwater-dependent area under such a climatic situation (this emphasize is re-highlighted on Line 73 – 76).

Of course, this method is also applicable in other areas with similar characteristics, especially due to the suggestion by one of the referees to add a new analysis and a new discussion section (results section 3.2, Figure 7, Line 363 – 382, and discussion section 4.2, Line 517 – 521). Direct-quoting the last part of the specific comments from referee #1, 'This should also be accompanied by a more detailed discussion of the assumptions (are your recharge conclusions robust wrt model assumptions and chosen parameter values?). This should help clarify whether the small changes in recharge are related to the physical characteristics of the basin or to the way the model calculates recharge. Such an analysis can also increase the scientific value of the paper beyond the case study.'. We agree that the suggested added analysis and discussion would increase the scientific value of the paper 'beyond the case study', therefore the abovementioned parts are added to serve this purpose.

We believe we have addressed all the issues accordingly. Looking forward to your and the referee's response to this. Thank you very much.

Rebuttal documents for the manuscript entitled:

"The impact of future climate projections and anthropogenic activities on basin-scale groundwater availability", submitted to the Hydrology and Earth System Sciences Journal

Referee #1 comments:

The paper quantifies the impact of future climate change and changes in groundwater pumping on groundwater resources in the Bandung groundwater basin, Indonesia. This is done by driving a surface water and groundwater model with CMIP6 climate projections and various groundwater abstraction scenarios. Results show that groundwater abstraction has a larger impact on groundwater levels/storage than climate-induced changes in groundwater recharge.

The paper tackles an important and relevant topic and is generally well written. The following comments identify several points that deserve attention.

Response to Referee #1:

We would like to thank you for the effort the referee provided to review our manuscript. We also thank you for the positive brief summary of our paper that you mentioned in the general comments of the review.

Referee #1 specific comment no. 1:

Based on the introduction, novelty of the paper seems to be largely limited to the case study, since the literature review shows that very similar methodology has been used before with similar conclusions (groundwater abstractions more important than climate change). To justify publication in HESS the authors should strengthen the novelty description of their work in the introduction. Otherwise, this paper may be better suited for a case-study oriented journal.

Response to Referee #1 specific comment no. 1:

Thank you for your feedback. We agree that the previous version of the introduction was indeed limited, as it refers unbalancely to one school of thought on the climate change impact on groundwater resources. The old manuscript focuses only on studies that agrees with the result of our analysis, however, the referee's comment gives us another idea and perspective that studies with diverging results are also important to be mentioned. By addressing those studies in our revised introduction, the spatial variability on the matter of climatic and anthropogenic factors' influence to the groundwater availability is explicitly mentioned, therefore strengthen the importance of studies on this topic on various spatial scales; globally, regionally, and even locally. We believe our new version of the manuscript has responded your comments sufficiently.

Referee #1 specific comment no. 2:

One of the main conclusions is that recharge is not significantly affected by climate change. I think this result should be more extensively explained and discussed. For example, it would be useful to provide more details about how recharge is calculated. I understand the modeling has been detailed in previous papers, but the recharge calculations are central to the current paper, so they deserve special attention. This could be accompanied with more detailed results e.g. time-series of computed soil water balances and groundwater tables, to more clearly demonstrate where the increased rainfall ends up and why. This should also be accompanied by a more detailed discussion of the assumptions (are your recharge conclusions robust wrt model assumptions and chosen parameter values?). This should help clarify whether the small changes in recharge are related to the physical characteristics of the basin or to the way the model calculates recharge. Such an analysis can also increase the scientific value of the paper beyond the case study.

Response to Referee #1 specific comment no. 2:

Thank you very much for this great suggestion. In the current version of the manuscript, we have provided the clarity of this matter. First, we added the recharge calculation scheme within the used hydrological model in Section 2.4.2: Wflow_sbm model setup, particularly about *MaxLeakage*, the influencing model parameter that regulate the simulated groundwater recharge output. Then, we discuss the recharge generation process in both the results (3.2) and discussion (4.2) sections. We also, based on your suggestion, accompany the simulated groundwater recharge with more detailed results; we do it with the simulated river discharge and actual evaporation in the results section 3.2, shown in the new Figure 7. In line 363 – 382, it is mentioned that the magnitude of groundwater recharge is relatively constant despite the increase in precipitation, as the rise of the forcing influx is reflected more so by the increase of river discharge as the outflow, and less so by the changes in groundwater storage, hence relatively constant groundwater recharge. The fact that both the median and the extreme values of the river discharge increases support the notion that groundwater recharge is less affected by changes in climate variables (the quantitative values of this analysis is presented in the new Figure 7a and 7b). Your comment on the last sentence that suggests that this would increase the scientific value of the paper beyond the case study is also mentioned in the discussion section (4.2) Line 517 – 521, showing that both the hydrological modelling scheme and the basin's physical characteristics, especially related to soil moisture capacity, plays an important role in recharge generation, regardless of the changes in the climate variables. Again, this is such a great suggestion, and we believe that the quality of our manuscript is gretly enhanced after incorporating this particular comment into it.

Referee #1 specific comment no. 3:

I'm missing an aquifer water balance, this can be very useful to put the recharge and pumping values in perspective and to assess sustainability of the system under different scenarios.

Response to Referee #1 specific comment no. 3:

Thank you for another great suggestion. We agree that the addition of aquifer water balance would be very useful to visualize the propagation of these two variables (recharge and abstraction) to the storage projection. A new figure on this visualization with its description is available on Section 3.4 (Groundwater storage projection). We also discuss it in Section 4.2 (Impact assessment on future groundwater level projection), as we can see from the additional figure 9, that these times (between year 2020 and 2025) are the crucial time as volumetric-wise, the annual groundwater abstraction is estimated to be at the cross-section with the total annual recharge. Once the groundwater recharge volume as the main inflow has been breached, it might be way more difficult to restore the basin's groundwater storage condition, despite other fluxes involved (surface – groundwater interaction, for example).

Referee #1 specific comment no. 4:

title: it's not the projections that will impact water availability, so better to change "future climate projections" to "future climate change" or "future groundwater recharge" (unless you actually mean that the projections will lead to decisions that will impact groundwater availability). Also, I would suggest to change "anthropogenic activities" to something more specific like "groundwater abstractions" or similar.

Response to Referee #1 specific comment no. 4:

Thank you for your suggestion. We agree to change the title of our manuscript so it can describe the content more precisely to: 'The impact of future changes in climate variables and groundwater abstraction on basin-scale groundwater availability'.

Referee #1 specific comment no. 5:

line 101: to what extent is the aquitard spatially continuous?

Response to Referee #1 specific comment no. 5:

Thank you for your question. Despite not being actually 100% continuous within the whole groundwater basin boundary, such a setup (of continuous thin aquitard layer) shown in the conceptual model (Figure 1b) is the dominant setting indicated in most of the data that we have. We derive our interpretation on the aquifer layering based on borehole data distributed across the basin, collated in previous studies (Rahiem, 2020) – mentioned in Section 2.4.3. We reported in our previous studies that these data are not uniformly distributed, especially in highly elevated area. To fill in the gap of knowledge on the aquifer layering profile, we derive the conceptual model parsimoniously, and verify the assumption with the simulation results. This way, not only we tackle the data sparsity by comparing the simulation results both qualitatively and quantitatively (reported in our previous studies), but also we verify our model assumption to define the aquitard as a continuous layer within the calculation. Despite minority of the borehole data (~ 5%) does not show the aquitard layer in its soil stratigraphy, we believe our approach to simplify the model setup in this setting is optimum in representing the basin subsurface physical characteristics. We do not explain all of this in the manuscript, as you can see, because it is pretty extensive (this will take at least 15 to 20 additional lines), and it has been mentioned and discussed in our previous papers that we refer to in the manuscript.

Referee #1 specific comment no. 6:

figure 1a: can you explain how the basin was delineated? is it based on topography?

Response to Referee #1 specific comment no. 6:

Thank you for your question. In answering this (and the upcoming) question, first we apologize if we refer the answer a lot to our previous studies. To provide a short answer to the question, the groundwater basin shown in Figure 1a is delineated by considering two sources: (1) the official Indonesia government document on the boundaries of groundwater basin and (2) surface topography MERIT-DEM (Yamazaki, et al, 2017). In the first document, it is stated that the groundwater basin is delineated based on the subsurface data that are collated by the government. Further information on the raw data used to delineate the basin was, unfortunately, not available/accessible. To verify its accuracy, we compare the groundwater basin delineation with the surface catchment area, and they are very similar (again, the comparison of these two boundaries was shown in our previous paper). To provide some minor adjustment considering some context and logical alignment between the surface and the subsurface basin, we incorporate the surface topography from the MERIT-DEM products to fine-tune the final groundwater basin boundary that we use in this study. MERIT-DEM is chosen as the benchmark for the sake of consistency, as we also use MERIT-DEM in our hydrological simulation using Wflow_sbm model.

Referee #1 specific comment no. 7:

line 150: one-way coupling is justified if water tables are relatively far below land surface, is that the case here?

Response to Referee #1 specific comment no. 7:

Thank you for your question. In our test basin, the surface elevation variation is large, causing the water table to also have a large variation. The lowest point in the basin is found at around 600 meters above sea level, while the highest point at around 2400 meters. Therefore, in the mountainuous area, the water table is relatively far below the surface up to several tens of meters. On the other hand, in the lower elevated area, the water table varies from a couple of ten meters depth to close to the surface water as the boundaries (especially around the river). We address this limitation in our discussion, Line 517 – 521.

Referee #1 specific comment no. 8:

line 168: "in each period"

Response to Referee #1 specific comment no. 8:

Thank you for your suggestion. We have adjusted the manuscript accordingly.

Referee #1 specific comment no. 9:

figure 2 could perhaps be simplified by only showing the workflow once but then with different climate and abstraction forcing

Response to Referee #1 specific comment no. 9:

Thank you for your suggestion. We like the idea of simplifying the workflow, however, we also want to show the 'repetitive' nature of the simulation. By visualizing the workflow 'twice', we want to deliver the message that the model during the baseline simulation (the left part of the figure) has been calibrated and even reported in our previous studies, and the current manuscript is the application of the one-way coupled scheme to project the future groundwater availability under various scenario (the right part of the figure). We think that this message is harder to convey should we simplify the figure to just one workflow, therefore suggesting to keep the figure as it is. We hope you understand and content with the reasoning.

Referee #1 specific comment no. 10:

line 181: the method for estimating potential ET is based on temperature and radiation and thus ignores potential changes in humidity and wind - can you justify this simplification or discuss its impact? Also, this method was apparently not developed for computing potential ET, so why is it applicable for this purpose?

Response to Referee #1 specific comment no. 10:

Thank you for your question. Indeed, considering the data limitation and uncertainties, we chose to involve methods in estimating PET to be simple. The justification is shown in the referred paper (de Bruin, et al. 2016), that the method is mostly confined to cases without local advection, which means that the local evaporation is not influenced as much by the moving moisture. As our study location is located in a 'closed system' (the Bandung area is basically a 'bowl' surrounded by mountain), the moisture taken away by the wind and humidity difference is insignificant, therefore allowing for the methods application in the area. In the same paper, it was also mentioned that the terminology potential evapotranspiration itself consists of multiple components, and our understanding over the term has evolved from time to time. Depending on the context of the usage, the methods can also be used to estimate the 'so-called potential evapotranspiration'. There have been a lot of studies that use this method to estimate potential evapotranspiration too (Oudin, et al. 2005, Imhoff, et al. 2020, Gebremehdin, et al. 2022).

Referee #1 specific comment no. 11:

line 196: did you check that the surface shortwave radiation from MRI-ESM2-0 is consistent with that from GFDL-ESM4? Or alternatively explain why this comparison is not needed.

Response to Referee #1 specific comment no. 11:

Thank you for your question. We did check the surface shortwave radiation from MRI-ESM2-0 and compared them with ones from GFDL-ESM4. There are minor differences (obviously), in which the MRI-ESM2-0 projection results in a slightly larger changes in future radiation. However, the effect of this differences in the following calculation is insignificant. First, the radiation is one variable among the other variables to compute the potential evapotranspiration (PET). Then, the PET is used not as a direct number/input in a calculation, but as the upper limit of the calculated actual evaporation. The actual evaporation then influences the magnitude of available water to flow as runoff or to infiltrate and further recharge the groundwater. These multi-step computation process dissipates the influence initiated by the minor differences between the MRI-ESM2-0 and the GFDL-ESM4 projection radiation products.

Referee #1 specific comment no. 12:

line 230: are there any rain gauges in the area to check the assumption of treating CHIRPS as ground truth?
Response to Referee #1 specific comment no. 12:

Thank you for your question. Yes, there are 11 rainfall stations within and around the area. We have discussed and reported our analysis (in our previous study) on the rainfall estimates in the area by comparing numerous rainfall estimation methods: rainfall station measurement, satellite-products, ground-corrected remote-sensing products, and interpolated rain gauge-based estimates. While the rainfall estimates measured by the rainfall stations are supposed to be the most accurate, they represent point measurement as opposed to areal estimates. Therefore, to derive the best estimate of the areal rainfall, we applied an uncertainty quantification method (Extended Triple Collocation method) in our previous study and conclude that CHIRPS could be used as the best estimate in representing the ground-truth areal rainfall. We did not provide a lengthy discussion on this matter in our manuscript, but it is shortly mentioned in Line 264-265, Section 2.4.2: Wflow_sbm model setup.

Referee #1 specific comment no. 13:

line 250: river discharge I assume

Response to Referee #1 specific comment no. 13:

Thank you for your suggestion. We have adjusted the manuscript accordingly.

Referee #1 specific comment no. 14:

line 252-253: shouldn't you be using bias-corrected CMIP6 data for the historical period? Similar comment for figure 4: show the bias-corrected MRI-ESM2-0 for historical period instead of CHIRPS.

Response to Referee #1 specific comment no. 14:

Thank you for your suggestion. We intentionally use CHIRPS as we treat it to represent the ground-truth rainfall estimates. We also treat the following Figure 5 similarly: using ERA5 data as the ground-truth instead of bias-corrected MRI-ESM2-0 estimates. We would like to keep the consistency in all the figures, not only among these figures, but also with the Figure 2 (the one about the workflow mentioned above). In that figure, the workflow is 'repeated' between the left and the right part of the figure. The left part represents the simulation using the 'ground-truth' as the forcing and boundaries, while the right part represents the future projection. We also do not directly compare the 'ground-truth' with future projections in Figure 4 and 5, but apply the bias-correction beforehand, with the aims that these comparisons are referenced on the more similar (bias-corrected) yet consistent (with the previous figures) ground.

Referee #1 specific comment no. 15:

line 274: what are the values for the storage parameters? and what are the "river-related parameters? Do the latter overlap with parameters in the wflow model?

Response to Referee #1 specific comment no. 15:

Thank you for your question. The storage parameter includes the specific yield ($s_y$) and specific storage ($s_s$) of the unconfined and confined aquifer. The $s_y$ value is 0.2, and $s_s$ 8.7*10$^{-3}$ /m. Again, this value has been reported in our previous studies referred to in the manuscript. The river related parameters include river cell, riverbed hydraulic conductance, surface water elevation (as the head boundary to the groundwater), and the elevation of the bottom of the riverbed. We use different values as the cell size in the surface hydrology and groundwater flow model are different, but they represent the similar condition. For example, in the groundwater flow model, we could not specify the river width, as cells assigned as rivers have to have the whole cell treated as a stream. In reality, the cell width of 100 m is not always consistent with the actual river width, thus we adjust our parameterization in order to make them mathematically representative.

Referee #1 specific comment no. 16:

figure 5b: not clear what the extra horizontal lines are in this plot

Response to Referee #1 specific comment no. 16:

Thank you for your detail review. The one in the boxplot is the surface downwelling shortwave radiation, while the one at the top (also in boxplot, but very thin that they look like lines) is top of atmosphere incident shortwave radiation. We have revised the figure accordingly to make it clear by adding a dashed line, separating the two mentioned 'regions'.

Referee #1 specific comment no. 17:

line 374: and much smaller storage coefficient in the confined aquifer?

Response to Referee #1 specific comment no. 17:

Thank you for your suggestion. We agree with your analysis on this matter, thus we have revised the manuscript accordingly.

Referee #1 specific comment no. 18:

figure 7: make colorbar title and labels more readable.

Response to Referee #1 specific comment no. 18:

Thank you for your suggestion. We have revised the now Figure 8 (there is an additional Figure 7 for the hydrological variables accompanying the groundwater recharge) accordingly by enlarging the size of the colorbar title and labels.

Referee #1 specific comment no. 19:

line 416: do your simulations predict decreases in baseflow?

Response to Referee #1 specific comment no. 19:

We did not initially collect the baseflow simulation output intentionally. Thanks to the input from your previous suggestions of accompanying the groundwater recharge with other variables, we now have these values in hand. The groundwater level projections show that the least significant future groundwater table changes are found in the unconfined layer along the stream. This shows that the river baseflow is relatively constant. While this seems not to be a problem, the groundwater table in the unconfined storage actually depletes the most in the mountainous region, contributing to conserving the river baseflow due to the groundwater abstraction along the river downstream part. This is now added to the manuscript (shown above in the discussion part after the new figure showing the aquifer water balance components).

Referee #1 specific comment no. 20:

line 479: what do you mean by "pseudo water table"?

Response to Referee #1 specific comment no. 20:

Thank you for your question. As said in that particular section of the manuscript, 'In the one-way model coupling setup, groundwater recharge is fully controlled by the surface processes and the pseudo-water table'. This sentence refers to the computational scheme of the hydrological model Wflow_sbm, where the soil is considered as a bucket with a certain depth and divided into saturated and unsaturated zone. The top of the saturated store represents the pseudo water table. The formula and detail description of all the Wflow_sbm parameter is now available online in this paper:
van Verseveld, W. J., Weerts, A. H., Visser, M., Buitink, J., Imhoff, R. O., Boisgontier, H., Bouaziz, L., Eilander, D., Hegnauer, M., ten Velden, C., and Russell, B.: Wflow_sbm v0.7.3, a spatially distributed hydrological model: from global data to local applications, Geosci. Model Dev., 17, 3199–3234, https://doi.org/10.5194/gmd-17-3199-2024, 2024.

Referee #1 specific comment no. 21:

line 485: "In regions with higher margins between the groundwater recharge and soil capacity". Not clear, please clarify.

Response to Referee #1 specific comment no. 21:

Thank you for your question. In that section, we aimed to describe the circumstances where changes in climate variables would have a higher influence on the changes in groundwater recharge. According to the result in the test basin, the threshold for the water in the vadose zone to infiltrate as groundwater recharge is generally low; the vadose zone is relatively wet due to many factors: the surface processes, the continuous rainfall period, the soil characteristics, etc. Therefore, in regions with higher soil capacity and lower soil moisture, generally, there is a higher possibilities of increasing groundwater recharge due to changes in climate variables. We hope that this explanation provides more clarity.

Referee #1 specific comment no. 22:

check erroneous text on line 576.

Response to Referee #1 specific comment no. 22:

Thank you for your detail review. We have removed the text.

Rebuttal documents for the manuscript entitled:

"The impact of future climate projections and anthropogenic activities on basin-scale groundwater availability", submitted to the Hydrology and Earth System Sciences Journal

Referee #2 comments:

The manuscript deals with a very interesting theme not only for the scientific community but for the entire society, that of the comparison of the impact of projected climate forcing and anthropogenic activity on future groundwater status. I found the manuscript well written with a quite correct structure and I can say that someone can read it with pleasure. The study is quite interesting and actually, I have occupied with the same case in the past ending up with the same conclusions. For that reason, I deposit a few comments for the paper improvement.

Response to Referee #2:

We would like to thank you for the effort the referee provided to review our manuscript. We also thank you for the positive brief summary of our paper that you mentioned in the general comments of the review.

Referee #2 specific comment no. 1:

Lines 19-20: The authors indicate that there are many basins worldwide with over-exploited groundwater resources, but the references presented are only three (3). More references should be added to satisfy the word "worldwide".

Response to Referee #2 specific comment no. 1:

Thank you for your feedback. The references for the mentioned part are actually provided in the sentences after, and not in that particular sentence. We understand that to satisfy the word 'worldwide' is not an easy task, thus the entire paragraph is dedicated to supply these examples (including the references). Following the Line 19-20 as you mentioned, the example of basins with over-exploited groundwater are mentioned until Line 28 (Bangladesh, China, Brazil, and Spain were at least mentioned).

Referee #2 specific comment no. 2:

Chapter 2.1. No information is given about the water uses, the crop types, or the volumetric budget of the aquifers.

Response to Referee #2 specific comment no. 2:

Thank you for your feedback. The water uses in the study area are dominated by the domestic and industrial sectors. This information along with the volumetric budgets of the abstractions has been mentioned in Line 93 – 97. We provided the more important details only in this study following the aim. For the other detailing and quantifications on pre-baseline modeling variables, they are mentioned in our previous studies that we mentioned as references multiple times in the manuscript.

Referee #2 specific comment no. 3:

Figure 1b. The line of the cross-section AA' should be presented in Figure 1a.

Response to Referee #2 specific comment no. 3:

Thank you for your suggestion. The figure has been adapted accordingly to your comment.

Referee #2 specific comment no. 4:

Line 109 "In recent years, the groundwater situation has not been improving" This is a very vague phrase and not properly stated. What does the word "situation" mean? I can understand very well the meaning of this sentence, but it needs to be rephrased to be more specific eg situation of what? The quantity? The quality?

Response to Referee #2 specific comment no. 4:

Thank you for your suggestion. We agree to change the phrasing accordingly. In the newer version, we are more direct and states what the 'situation' means, practically.
* * *
Referee #2 specific comment no. 5:

Lines 238-250: I can understand that a full description of the hydrological model parameterization is reported in the previous studies (Rusli et al., 2023a, b), but a very brief report should be given here only for the most important parameters.

Response to Referee #2 specific comment no. 5:

Thank you for your suggestion. We have added an information on that section, especially on the most important parameter named *MaxLeakage* that controls the amount of groundwater recharge in the hydrological simulation, specifically in Section 2.4.2: Wflow_sbm model setup.
* * *
Referee #2 specific comment no. 6:

Lines 263-279: I can understand that a full description of the groundwater flow model parameterization is reported in the previous studies (Rusli et al., 2023a, b), but a very brief report should be given here only for the most important parameters the storage coefficient of the aquifers, the conductance of the hydraulic connection between the rives and the upper aquifer, the starting conditions, the period of simulation, the volumetric budget.

Response to Referee #2 specific comment no. 6:

Thank you for your suggestion. We have quantified all the parameters and model settings you mentioned in the manuscript in Section 2.4.3: Groundwater flow model setup.
* * *
Referee #2 specific comment no. 7:

Lines 317-318: How do you explain the fact that the surface radiation reveals a tendency of a slight reduction in the future? Is this a normal, and expected result? Please justify your answer with a reference to other studies.

Response to Referee #2 specific comment no. 7:

Thank you for your question. The projected surface radiation, in our opinion, vary spatially. While some areas are expected to have increasing values, some others might be predicted to have decreasing values. For example, the projected cloud covers changes over time in different area, therefore changing the radiation values. Thicker cloud covers would decrease the surface radiation, and vice versa. As per your suggestion to justify the decreasing values, here below are some references we found that support this numbers, where they found a decrease in future surface radiation projection.

- Ruosteenoja, K., Räisänen, P., Devraj, S., Garud, S. S., & Lindfors, A. V. (2019). Future Changes in Incident Surface Solar Radiation and Contributing Factors in India in CMIP5 Climate Model Simulations. Journal of Applied Meteorology and Climatology, 58(1), 19–35. https://www.jstor.org/stable/26679319
- Watanabe, S., K. Sudo, T. Nagashima, T. Takemura, H. Kawase, and T. Nozawa (2011), Future projections of surface UV-B in a changing climate, J. Geophys. Res., 116, D16118, doi:10.1029/2011JD015749.
- Martin Wild, Doris Folini, Florian Henschel, Natalie Fischer, Björn Müller. Projections of long-term changes in solar radiation based on CMIP5 climate models and their influence on energy yields of photovoltaic systems, Solar Energy, Volume 116, 2015, Pages 12-24, ISSN 0038-092X, https://doi.org/10.1016/j.solener.2015.03.039.

Referee #2 specific comment no. 8:

Figure 5.c. How do you explain the fact that the potential evapotranspiration reveals a reduction in the future, since the temperature increases? Furthermore why the potential evapotranspiration of RCP8.5 is lower than the one of the RCP4.5? Is this a normal, and expected result? Please justify your answer with a reference to other studies.

Response to Referee #2 specific comment no. 8:

Thank you for your question. The future potential evapotranspiration, as shown in Figure 5c, has a temporal variation according to its projection values. At some months, February to May and August to be specific, the PET actually increases and not decreases. It is 5 months among the 12 there are, so we would say that generalizing the PET value to be reduced is not representative to the results. About the comparison between the RCPs, they also vary for each month. For those periods where the PET of RCP8.5 is smaller, we would argue that the PET is controlled by other variables other than temperature (e.g. radiation). Therefore, despite the temperature increases, the projection of other variables also play an important role in determining the PET projection.

Referee #2 specific comment no. 9:

Chapter 3.3. The presentation of the results, although is very understandable and compact, is poor regarding both the size of the text and the use of the tables/graphs.
For example:

- In Lines 373-384 where the results of the the increasing groundwater abstraction scenario are presented, the results of 1) the RCP 4.5 scenario for the unconfined aquifer, 2) the RCP 8.5 scenario for both the two aquifers are missing. The phrase "Under the RCP8.5 scenario, the numbers are also concerning for the unconfined aquifers, as the groundwater table is projected to dwindle to up to 3.38 m and 3.40 m in the long run under the RCP4.5 and RCP8.5 scenarios, respectively" is not enough.
- The maps with the hydraulic head changes is a good and very useful choice but the results of the RCP 8.5 scenario and that of the other groundwater abstraction scenarios are not presented. I understand that many maps are needed for that reason, but the authors can focus on the presentation of the worst-case scenarios. There is no use in presenting the maps of Figures 7.a,b,c since the drawdown is not higher than one meter. I propose the authors find an extra way of representing the decline of the groundwater levels in relation to the time e.g. a line for each climatic and groundwater abstraction scenario (6 scenarios * 2 aquifers = 12 lines -like Fig.8) showing the decline of the groundwater level for that cell of aquifers that reveals the maximum drawdown. The same could be done for the drawdown area.

Response to Referee #2 specific comment no. 9:

Thank you for your question. Indeed, there are a lot of numbers to unpack in Section 3.3. We were thinking to report all the numbers, but that would be tedious, not only to write, but also to be read by the audience. Therefore, we focus on the extreme values in the text, and after discussing further until Line 406, we refer to Table 3 that summarizes all the values in one table, with the aim to make it easier to read. Meanwhile, about the figure where the drawdown is shown, we opted to show the Figure a,b, and c, as they represent the unconfined aquifer in the short-, mid-, and long-term scenario. We very much want to have a representation in both layers of aquifers, and all three future checkpoints. We also put the results with the main aim to show the spatial distribution of the groundwater head change, and not the magnitude (although this is also interesting to see), because the magnitude is revealed in Table 3 (and Table 3 cannot visualize the spatial distribution of the changes).

Referee #2 specific comment no. 10:

Lines 384-385: Why do you use the phrase "as expected"? It is not so obvious to the reader because the groundwater table decline was not presented in chapter 2.1. This result should be highlighted more and it should be made clear that even though the withdrawals do not increase the level will continue to fall and it should be explained why.

Response to Referee #2 specific comment no. 10:

Thank you for your detailed comment. Indeed, we wrote that with our three previous papers in mind, so the word 'as expected' was written. We agree that to the reader, it is not so obvious, and that phrase should be removed. We have revised the part accordingly.

---

## Author Response (AR2)

Rebuttal documents for the manuscript no. HESS-2024-26 entitled:

"The impact of future changes in climate variables and groundwater abstraction on basin-scale groundwater availability", submitted to the Hydrology and Earth System Sciences Journal

Editor comments:

Considering the reviewers' feedback, I recommend proceeding with the publication of the manuscript following minor revisions.

Response to Editor:

We sincerely thank you for the effort you have allocated to manage the reviewing and open discussion process of our manuscript. We see that one reviewer has suggested to accept the manuscript as it is, while the other one suggested to accept subject to minor revision. We have responded to both reviewers, while also accommodating the comments from the latter in our latest revision. All the issues have been addressed accordingly. Looking forward to your and the referee's response to this. Thank you very much.

sRebuttal documents for the manuscript no. HESS-2024-26 entitled:

"The impact of future changes in climate variables and groundwater abstraction on basin-scale groundwater availability", submitted to the Hydrology and Earth System Sciences Journal

Referee #1 comments:

I thank the authors for revising their manuscript. I believe the novelty is now more clearly articulated.

Response to Referee #1:

We would like to thank you for the effort the referee provided to review our manuscript. Indeed, thanks to your previous comments and suggestions, we believe the overall quality of the manuscript, especially how the novelty is articulated is now improved. Below we responded to your further comments individually.

Referee #1 specific comment no. 1:

Regarding the added details on recharge calculation, the sentence added on lines 262-265 is not entirely clear. How does the "maxleakage" parameter enter the calculations? Can you give the equation? Also, you seem to equate groundwater recharge with vertical groundwater movement between shallow and deep aquifers, which is confusing.

Response to Referee #1 specific comment no. 1:

Thank you for your feedback. Regarding the 'MaxLeakage' parameter, we agree to add more information, and to that extent, quoting the definition of the parameter from the model documentation itself. In the wflow_sbm model, MaxLeakage is usually only used for linking to a dedicated groundwater model, where it represents the water that 'lost' to the model. Normally set to zero in all other cases, when the MaxLeakage is set to be higher than zero, the simulated water is treated to be lost from the saturated zone (and runs out of the model). As wflow_sbm model only take into account the first couple of meters of soil below the surface level, the water that leaves the saturated zone is then treated as the groundwater recharge. We hope this explanation makes things clearer and easier to understand. Additionally, this description is now added in subsection 2.4.2 'Wflow_sbm model setup', paragraph 2, line 258 to 263 (clean version).

Referee #1 specific comment no. 2:

Since parameter "maxleakage" controls recharge in the model, the question arises whether this parameter can change. The assumption here seems to be that it is constant. But what if it changes? Is this plausible or not? And would that affect the recharge projections in any way? Some additional discussion on this seems relevant, for example around lines 534-545 where assumption of one-way model coupling is discussed. And can you explicitly state there whether it may be useful to revisit these assumptions in future work, and if not, why you think these assumptions are inconsequential for your conclusions.

Response to Referee #1 specific comment no. 2:

Thank you very much for this great suggestion. Indeed, when we set the model up, we started with the assumption that the MaxLeakage parameter remains constant during the simulation period. Your point is absolutely valid, as soil characteristics might evolve, albeit in a much longer time scale than the surface features. In our opinion, changes in soil characteristics would influence the MaxLeakage parameter, therefore also impacting the simulated projected groundwater recharge. While these changes are gradual, they are likely to affect recharge over time, potentially causing significant deviations from current projections. However, the tendency of how soil characteristics evolve long term, particularly regarding groundwater recharge generation, remains uncertain. Whereas, accurate future projections require constant soil monitoring and modeling. This further highlights the importance of data, especially soil-related data and hydrological information as the benchmark for model calibration and verification.

Consistent data assimilation that are validated through updated observation and simulation would decrease the uncertainties, making it possible to make the MaxLeakage parameter setup dynamic over time. Again, thank you again for such an interesting and critical question. We have added this to the discussion section as your suggested in subsection 4.2 'Impact assessment on future groundwater level projections', paragraph 3, line 534 to 545 (clean version).

Referee #1 specific comment no. 3:

Line 265: suggest to say "river discharge" instead of "discharge"

Response to Referee #1 specific comment no. 3:
Thank you for your suggestion. It has been adjusted accordingly.

Referee #1 specific comment no. 4:

Figure 9: what do you mean by "aquifer water balance of groundwater recharge and abstraction"? Is it abstraction minus recharge?

Response to Referee #1 specific comment no. 4:

Thank you for your question. We basically aim to plot and show the development of the two variables overtime: while the recharge (as impacted by the changes in climatic variables) remains relatively consistent, the groundwater abstraction (as the direct consequences of changes in anthropogenic factors) has a much higher change in relation to the aquifer water balance. Therefore, your deduction is right, and even one step further by looking at the difference between the propagation of the two variables (hence abstraction minus recharge, as you stated).

Rebuttal documents for the manuscript no. HESS-2024-26 entitled:

"The impact of future changes in climate variables and groundwater abstraction on basin-scale groundwater availability", submitted to the Hydrology and Earth System Sciences Journal

Referee #2 comments:

Accepted as is. No suggestions, major comments have been incorporated in the revised manuscript.

Response to Referee #2:

We would like to thank you for the effort the referee provided to review our manuscript. We believe the inclusion of response and suggestions from the previous round of reviews have improved the manuscript, thanks to your excellent review.